# Understanding CRY2 interactions for optical control of intracellular signaling

Liting Duan[1], Jen Hope[1], Qunxiang Ong[1], Hsin-Ya Lou[1], Namdoo Kim[2,3,4], Comfrey McCarthy[5], Victor Acero[6], Michael Z. Lin [2,3,4] & Bianxiao Cui[1]

*Arabidopsis* cryptochrome 2 (CRY2) can simultaneously undergo light-dependent CRY2–CRY2 homo-oligomerization and CRY2–CIB1 hetero-dimerization, both of which have been widely used to optically control intracellular processes. Applications using CRY2–CIB1 interaction desire minimal CRY2 homo-oligomerization to avoid unintended complications, while those utilizing CRY2–CRY2 interaction prefer robust homo-oligomerization. However, selecting the type of CRY2 interaction has not been possible as the molecular mechanisms underlying CRY2 interactions are unknown. Here we report CRY2–CIB1 and CRY2–CRY2 interactions are governed by well-separated protein interfaces at the two termini of CRY2. N-terminal charges are critical for CRY2–CIB1 interaction. Moreover, two C-terminal charges impact CRY2 homo-oligomerization, with positive charges facilitating oligomerization and negative charges inhibiting it. By engineering C-terminal charges, we develop CRY2high and CRY2low with elevated or suppressed oligomerization respectively, which we use to tune the levels of Raf/MEK/ERK signaling. These results contribute to our understanding of the mechanisms underlying light-induced CRY2 interactions and enhance the controllability of CRY2-based optogenetic systems.

[1] Department of Chemistry, Stanford University, Stanford, California 94305, USA. [2] Department of Bioengineering, Stanford University, Stanford, California 94305, USA. [3] Department of Pediatrics, Stanford University, Stanford, California 94305, USA. [4] Department of Neurobiology, Stanford University, Stanford, California 94305, USA. [5] Department of Psychology, Northeastern University, Boston, Massachusetts 02115, USA. [6] Department of Engineering Science, Pennsylvania State University, State College, Pennsylvania 16802, USA. Correspondence and requests for materials should be addressed to M.Z.L. (email: mzlin@stanford.edu) or to B.C. (email: bcui@stanford.edu)

Optical control of protein activity is a promising approach for studying molecular mechanisms of signal transduction with high spatiotemporal resolution in living cells. Optical approaches use photosensory domains that undergo light-induced protein–protein hetero-dimerization, such as the LOV (light-oxygen-voltage) domain of *Avena sativa* phototropin 1, *Arabidopsis* phytochrome B and *Arabidopsis* cryptochrome 2 (CRY2), protein–protein homo-interaction, such as the LOV domain of *Vaucheria frigida* aureochrome1, *Synechocystis* phytochrome 1 and *Arabidopsis* CRY2, or protein–protein dissociation, such as *Arabidopsis* UVR8, LOVTRAP, and pdDronpa[1–19]. Among photosensory proteins, *Arabidopsis* CRY2 is unusual as it exhibits both homo-oligomerization and hetero-dimerization in response to blue light[20]. Cryptochromes, photoreceptors found in both plants and animals, contain a N-terminal photolyase homology region (PHR) that binds to a flavin adenine dinucleotide molecule as the blue light-absorbing cofactor[21]. In *Arabidopsis*, CIB1 (cryptochrome-interacting basic-helix-loop-helix) was identified to interact with CRY2 in a blue light-specific manner to promote floral initiation[12]. This light-induced CRY2-CIB1 hetero-dimerization, mapped to the PHR domain of CRY2, was then used for optical control of protein-protein interactions[13]. The CRY2–CIB1 system has been successfully utilized for optical control of intracellular signaling[22–25], lipid metabolism[26], gene expression[27–30] and organelle transport[31], and has been found to function robustly in a variety of organisms without requiring an exogenous cofactor[27, 29, 30].

In addition to binding CIB1, CRY2 also undergoes homo-oligomerization upon blue light stimulation, which was initially observed in plant cells[32]. The light-dependent homo-oligomerization of CRY2 was first employed as an optogenetic method to activate Wnt and Rho-family GTPase pathways[33]. Henceforth, optical strategies based on CRY2 oligomerization have been used to activate a wide range of signaling molecules, including Raf[34, 35], receptor tyrosine kinases[36, 37], and actin-nucleating proteins[38]. Moreover, light-induced CRY2 oligomerization has also been utilized for optical sequestration or inhibition of target proteins by inducing the formation of large clusters[33, 34]. This has been exploited to inhibit diverse proteins regulating the cytoskeleton, lipid signaling, the cell cycle, and endocytosis[38, 39]. Recently, CRY2olig, a CRY2 mutant (CRY2 (E490G)) with increased oligomerization capacity has been shown to improve CRY2-dependent protein activation or sequestration[38].

On the other hand, CRY2 oligomerization poses complications for applications that use CRY2-CIB1 hetero-dimerization. The CRY2–CIB1 system has been used in a number of studies to achieve protein translocation and light-induced binding of two different proteins[13, 19, 20, 23–28]. Previous studies have shown that CRY2 oligomerization and CRY2–CIB1 binding co-exist in the same systems[17, 36]. Therefore, unintended homo-interaction is

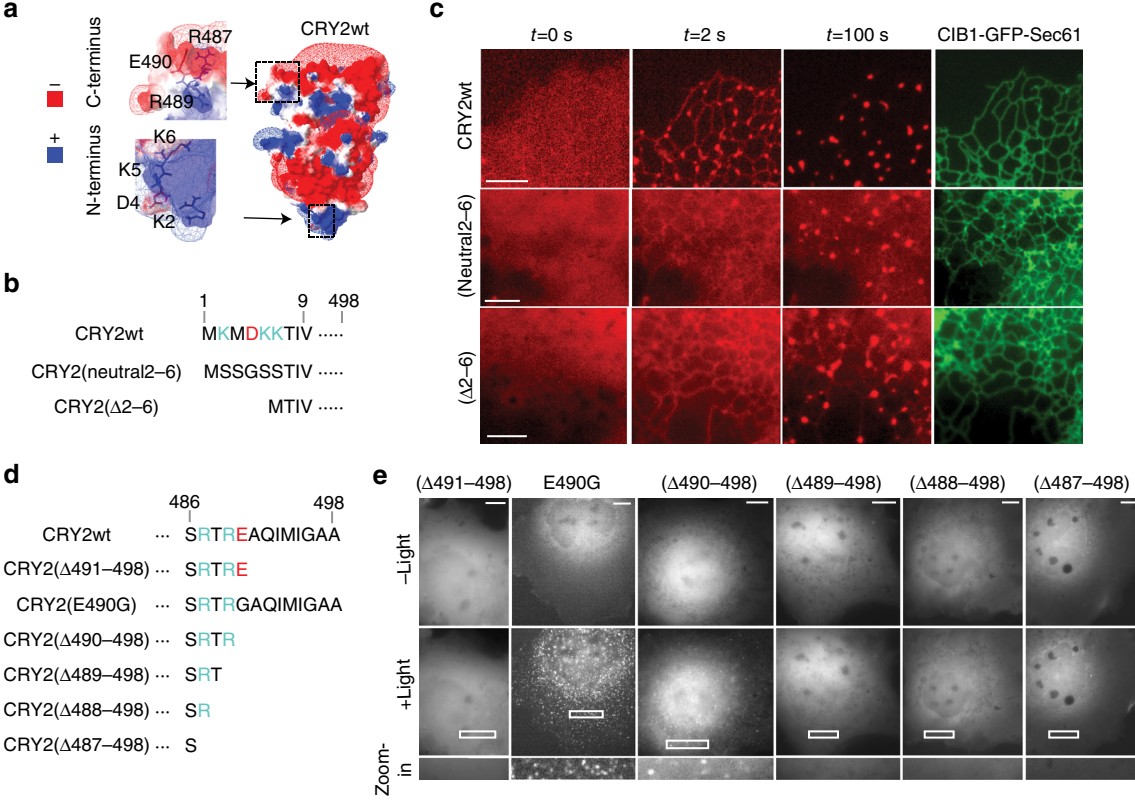

**Fig. 1** Electrostatic charges at N- and C-termini are important for CRY2–CIB1 and CRY2–CRY2 interactions, respectively. **a** Surface charge distribution of CRY2wt predicted by homology modeling. Charged residues are labeled in the insets of both the N-terminus and C-terminus. **b** Electrostatic charges at the N-terminus of CRY2 are removed by having charged amino acids replaced with neutral ones (CRY2(neutral2-6)) or deleted (CRY2(Δ2-6)). **c** The light-induced CRY2-CIB1 binding is much weaker for CRY2(neutral2-6) or CRY2(Δ2-6), as compared to CRY2wt. COS7 cells were co-transfected with CIB1-GFP-Sec61 and each mCh-tagged CRY2, respectively. Blue light was delivered at 2-s intervals for 100 s. **d** The amino acid sequences for the C-terminus of CRY2wt, CRY2(E490G) and each truncated CRY2. Truncated CRY2 variants, including CRY2(Δ491–498), CRY2(Δ490–498), CRY2 (Δ489-498), CRY2(Δ488–498) and CRY2(Δ487–498), were constructed by sequentially removing residues from CRY2(487–490). **e** Cytosolic CRY2 (E490G) and CRY2(Δ490–498) formed clusters upon light stimulation, while the other C-terminally truncated derivatives did not. COS7 cells were transfected with each mCh-tagged CRY2 and stimulated with blue light at 5-s intervals for 500 s. *Scale bars*, 5 μm (**c**), 10 μm (**e**)

induced within the same protein species, in addition to the hetero-interaction of the two different proteins. To date, there has been no report of CRY2 mutants with reduced oligomerization tendency. Modulation of CRY2 oligomerization is challenging due to the lack of knowledge about its underlying mechanisms. There is no crystal structure for either monomeric or oligomeric CRY2, so the molecular interfaces between CRY2–CRY2 and CRY2–CIB1 remain elusive.

In this work, we provide mechanistic insights into light-induced CRY2 interactions. We discovered that charged residues at the N-terminus of CRY2 are critical to light-induced CRY2–CIB1 dimerization. We also found that electrostatic charges at C-terminal residues 489 and 490 drastically affect light-induced CRY2 homo-oligomerization, with positive charges facilitating oligomerization and negative charges inhibiting it. On the basis of this principle, we engineered a CRY2 mutant with drastically enhanced homo-oligomerization ability, CRY2high, and a CRY2 derivative with significantly reduced homo-oligomerization, CRY2low. To further reduce the clustering capacity, CRY2low was fused with a large fluorescent protein, tandem dimeric Tomato (tdTom), which sterically hinders oligomer formation. CRY2 variants with different oligomerization propensities offer an additional layer of optical control for biological processes, as demonstrated using an opto-Raf system to tune the levels of Raf/MEK/ERK signaling. We envision that CRY2high will be a useful tool for optical strategies that require robust CRY2 oligomerization. Moreover, as the first reported CRY2 variant with reduced oligomerization, we expect that CRY2low-tdTom can greatly improve the specificity of CRY2-CIB1 based applications by diminishing unwanted protein oligomerization. Finally, the mechanistic insights into CRY2 interactions we provide here can be used as a guide to further engineer CRY2 systems to achieve optical control of biological processes.

## Results

**Charged residues at N- and C-termini are important for CRY2–CIB1 and CRY2–CRY2 interactions respectively.** We first investigated whether electrostatic interactions are involved in *Arabidopsis* CRY2 interactions. Electrostatic interaction is recognized as one of the main driving forces for protein interaction[40]. A previous study revealed that exchange of positively charged residues to neutral ones at a coiled coil epitope of mammalian CRY2 significantly reduces its affinity for the transcription factor mBMAL1[41]. Hereafter, CRY2 denotes the PHR domain of *Arabidopsis* CRY2 (amino acids 1–498), the region used as a light-inducible actuator. To examine charge distributions on CRY2 and the effects of mutations, we used Modeller-based homology modeling to generate three-dimensional models of CRY2 and CRY2 derivatives from the crystal structure of the closely related *Arabidopsis* CRY1 [42–44], and used the Swiss-PdbViewer program to construct electrostatic potential maps. The model for wild-type CRY2 (CRY2wt) indicates that the first 6 residues at the N-terminus, which are exposed to solvent, are strongly positively charged due to the presence of three lysine residues, Lys-2, Lys-5 and Lys-6 (Fig. 1a). To probe the functional role of these lysines in CRY2 interactions, we replaced them with neutral amino acids, forming CRY2(neutral2-6) or deleted them, forming CRY2($\Delta$2–6) (Fig. 1b). In CRY2(neutral2-6) or CRY2($\Delta$2–6), the N-terminus becomes neutral (Supplementary Fig. 1a).

We then tested whether CIB1-binding ability was preserved in these CRY2 variants. Light-induced recruitment of CRY2 onto the endoplasmic reticulum (ER) membrane via CRY2-CIB1 hetero-dimerization has been previously used to indicate CIB1-binding activity[20]. COS7 cells were transfected with mCherry(mCh)-tagged CRY2wt, CRY2(neutral2–6), or CRY2($\Delta$2–6) along with CIB1-GFP-Sec61, where Sec61 denotes the transmembrane domain of the ER-targeting protein Sec61[45] and targets the CIB1-GFP-Sec61 fusion protein to the ER membrane with CIB1 facing the cytosol. In this study, all intermittent blue light stimulation was delivered as a series of 200-ms pulses (9.7 W cm$^{-2}$). Before any light stimulation, CRY2wt, CRY2(neutral2–6), and CRY2($\Delta$2–6) were diffuse in the cytosol (Fig. 1c, Supplementary Fig. 1b, Supplementary Movie 1). Blue light pulses were then delivered at 2-s intervals. After one pulse of light at $t = 2$ s, CRY2wt was completely recruited to ER membrane and cytosolic CRY2wt was nearly depleted, confirming the strong CIB1-binding ability of CRY2wt. At $t = 100$ s, ER-bound CRY2wt aggregated into large clusters, which agrees with our previous report that membrane recruitment drastically enhances CRY2 oligomerization[20]. In contrast, for CRY2($\Delta$2–6) or CRY2(neutral2–6), after one pulse of blue light, a small portion was recruited to the ER membrane with a significant fraction remaining in the cytosol (quantitative analysis of CRY2-CIB1 dimerization is shown in Supplementary Fig. 1c). Even at $t = 100$ s, a large fraction of CRY2(neutral2–6) and CRY2($\Delta$2–6) molecules still remained in the cytosol, indicating reduced affinity for CIB1. On the other hand, CRY2(neutral1–9) and CRY2($\Delta$2–6) oligomerize similarly to CRY2wt, both in cytosol and on the ER membrane (Fig. 1c and Supplementary Fig. 1d). In summary, these results indicate that the CRY2 N-terminus is critically involved in CRY2-CIB1 binding, but does not obviously impact CRY2 homo-oligomerization.

We next focused on the charged residues near the C-terminus of CRY2. While the last 8 residues (491–498) are neutral, the preceding 4 residues (487–490) contain three charged amino acids, Arg-487, Arg-489 and Glu-490 (Fig. 1a). In CRY2olig (CRY2(E490G)), a previously reported CRY2 derivative with enhanced oligomerization, the negatively charged glutamate at residue 490 is replaced with the neutral glycine[38]. To probe the charge effect, we sequentially removed C-terminal residues by truncating CRY2 from CRY2($\Delta$491–498) to CRY2($\Delta$487–498) while preserving the blue light-sensing region (a.a. 1-485)[46] (Fig. 1d). Our molecular modeling reveals that, in CRY2wt and CRY2($\Delta$491–498), the C-terminal surfaces are predominantly negatively charged, with only a small region of positive charge at the side chain of Arg-489 (Fig. 1a, Supplementary Fig. 2a). In CRY2(E490G) (CRY2olig) and CRY2($\Delta$490–498), in which Glu-490 is mutated or removed, the C-terminus is predicted to present a mix of neutral and positive surfaces. Further truncation to CRY2($\Delta$489–498), CRY2($\Delta$488–498), and CRY2($\Delta$487–498) reverts back to a predominantly negatively charged C-terminus (Supplementary Fig. 2a). These models suggest that, in this region of the molecule (shown in the insets), the negative charge at Glu-490 is critical in counterbalancing the positive charge at Arg-489 in enhancing CRY2 oligomerization.

We first examined whether the truncations at the CRY2 C-terminus affect light-mediated CRY2–CIB1 dimerization. COS7 cells were co-transfected with mCh-tagged truncated CRY2 variants along with CIB1-GFP-Sec61 (Supplementary Fig. 3). After one 200-ms pulse of blue light, mCh-tagged CRY2 variants were completely recruited to ER membrane, suggesting that the CIB1-binding is preserved. Next, to characterize the effects of CRY2 C-terminal truncations on light-induced homo-oligomerization, COS7 cells were singly-transfected with mCh-tagged CRY2wt, CRY2(E490G) (CRY2olig), or each truncated CRY2 mutant, respectively. Before blue light stimulation, all CRY2 variants showed diffuse cytosolic distributions (Fig. 1e and Supplementary Movie 2). Transfected cells were stimulated by light at 5-s intervals for 500 s in total. While CRY2wt and CRY2($\Delta$491–498) showed no discernable protein aggregation, CRY2

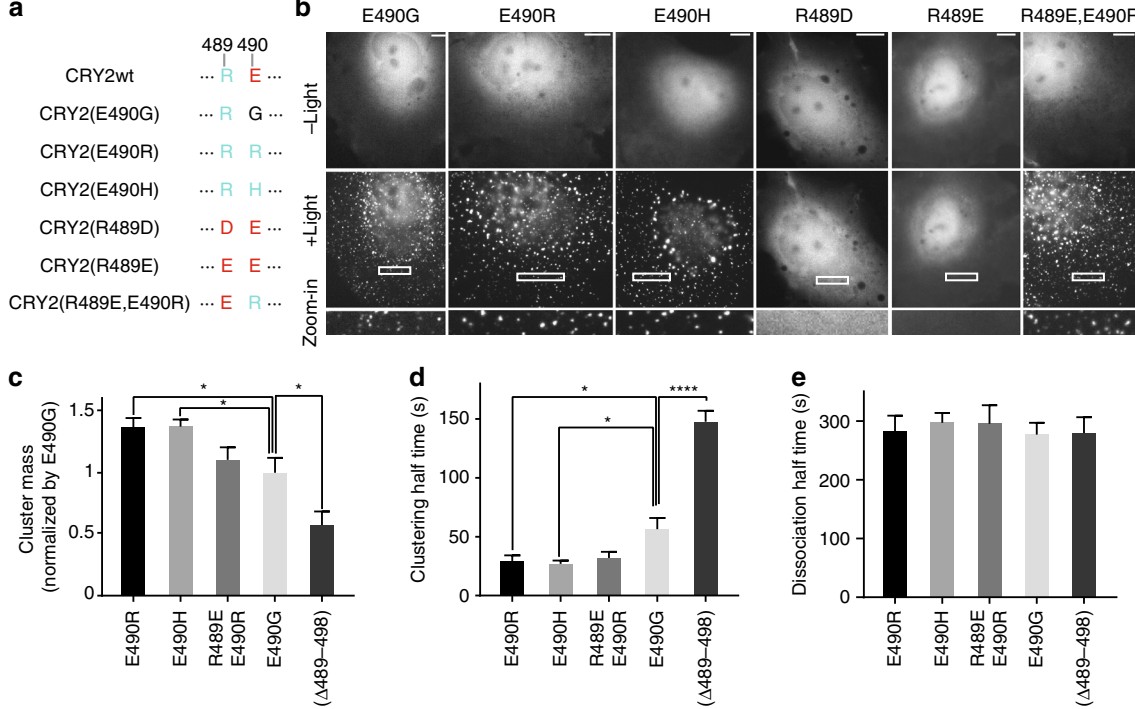

**Fig. 2** Double-positive charges at residues 489 and 490 enhance CRY2 homo-oligomerization. **a** A series of mutations were introduced at positions 489 and 490 of CRY2. **b** Cytosolic CRY2(R489D) and CRY2(R489E) did not form noticeable clusters, while the other mutants oligomerized into clusters drastically. Blue light was delivered at 5-s intervals for a total of 500 s. **c** The cluster mass was quantified using a custom Matlab program and normalized against that of CRY2(E490G). **d** Half-maximal clustering time of each CRY2 variant. **e** Dissociation half time of each CRY2 variant. Results are presented as means ± s.e.m. ($n = 13, 12, 14, 21, 14$). Results were analyzed using one-way ANOVA with Dunnett's post hoc test. (*$P < 0.05$, ****$P = 0.0001$). Scale bars, 10 μm

(E490G), in which the negative charge at residue 490 is neutralized, showed many noticeable clusters under identical conditions, consistent with previous observations[38]. CRY2 (Δ490–498), in which Glu-490 is deleted, also exhibited very clear aggregation. These results suggest that negative charge at residue 490 inhibits aggregation. Additional removal of the adjacent residue, the positively charged Arg-489, in CRY2 (Δ489–498) then completely abolished aggregation, indicating that Arg-489 can greatly enhance CRY2 aggregation. The further truncated derivatives CRY2(Δ488–498) and CRY2(Δ487–498) showed no cluster formation. These results indicate that light-induced CRY2 oligomerization is suppressed by Glu-490, but enhanced by Arg-489.

Taken together, the above results indicate that distinct regions of CRY2 are involved in its interactions with CIB1 or CRY2. As our aim is to tune CRY2 homo-oligomerization in this report, we focus on the role of C-terminal charges in CRY2-CRY2 interactions in the following sections.

**C-terminal positive charges enhance CRY2 homo-oligomerization.** After demonstrating that charged residues 489 and 490 play critical roles in CRY2 homo-oligomerization, we next investigated how positive or negative charges at the two locations affected CRY2 clustering. To this end, we mutated residues 489 and 490 of CRY2wt (Fig. 2a), producing two mutants with two positively charged amino acids, CRY2(E490R) and CRY2(E490H); two mutants with two negative charges, CRY2(R489D) and CRY2(R489E); and one mutant in which the positive and negative charges of CRY2wt are reversed, CRY2 (R489E,E490R). Surface charge modelling confirmed that CRY2 (E490R) and CRY2(E490H) have more positive charge at the

C-terminus, while CRY2(R489D) and CRY2(R489E) have more negative charge (Supplementary Fig. 2b). Each of these variants along with CRY2(E490G) was fused to mCh and expressed in the COS7 cell line. Upon blue light stimulation, the two derivatives with double negative charges, CRY2(R489D) and CRY2(R489E) did not form any visible clusters, similar to CRY2wt, while CRY2 (E490G) and the two mutants with double positive charges, CRY2 (E490R) and CRY2(E490H) showed dramatic protein aggregation (Fig. 2b and Supplementary Movie 3). Similarly, CRY2(E490K), possessing double positively charged residues at C-terminus, also clustered drastically in cytosol (Supplementary Fig. 4). It is worth noting that reversing the charge order at 489 and 490 (CRY2 (R489E, E490R)) induced clear CRY2 clustering, unlike CRY2wt, suggesting that the positive charge at 490 is more effective than at 489 in promoting CRY2 aggregation.

To examine the effect of C-terminal positive charges on CRY2 oligomerization, we quantified the cluster mass, clustering half time, and dissociation half time for CRY2 mutants that exhibit light-induced aggregation in cytosol (Fig. 2c–e). For this purpose, CRY2 clustering was induced by a single 1-s pulse of blue light (9.7 W cm$^{-2}$). All mutants tested showed similar dissociation half - times. Cluster mass was quantified in Matlab and normal-ized to cells expressing CRY2(E490G) (see methods for details). CRY2 variants with double positively charged residues, CRY2 (E490R) and CRY2(E490H), possessed greater cluster masses (both 0.29) and more rapid clustering (half time 29 and 27 s) than other mutants, including CRY2(E490G) (cluster mass 0.17, half time 56 s). Though bearing different positively charged amino acids at the 490 position, CRY2(E490R) and CRY2(E490H) showed very similar clustering capacities, which further supports the critical role of the electrostatic effect. We note that CRY2 (Δ490–498) showed lower cluster mass (0.10) and slower binding

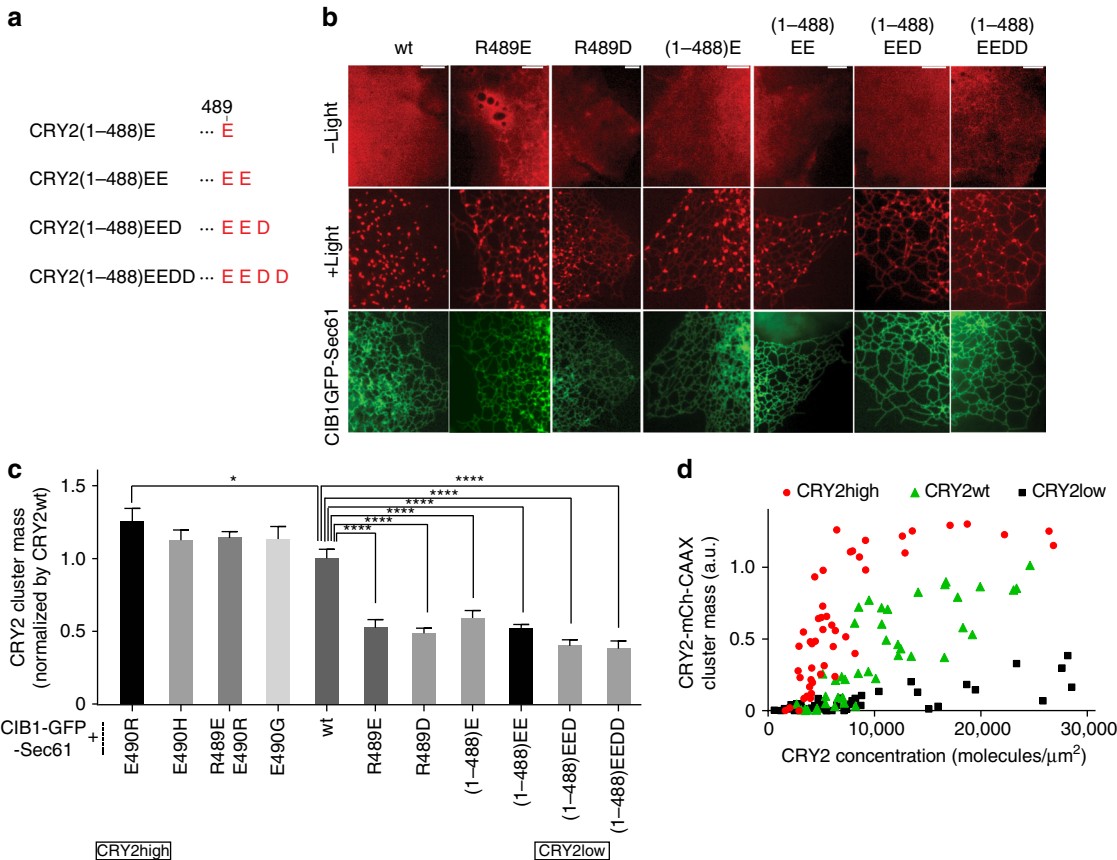

**Fig. 3** Negative charges at the C-terminus of CRY2 inhibit CRY2 homo-oligomerization. COS7 cells were co-transfected with CIB1-GFP-Sec61 and each mCh-tagged CRY2, respectively. Blue light of 200-ms duration was delivered at 2-s intervals for 20 s (**a–c**) or one 100-ms pulse of blue light was delivered (**d**). **a** Increasing numbers of negative charges were introduced to the C-terminus of CRY2(1–488). **b** After recruitment to the ER membrane via CRY2–CIB1 hetero-dimerization, CRY2 derivatives with more C-terminal negative charges formed fewer clusters as compared to CRY2wt. **c** The cluster mass of each CRY2 mutant on ER membrane was separately quantified and then normalized to that of CRY2wt. Results are presented as means ± s.e.m. (*n* = 13, 12, 15, 14, 16, 12, 11, 13, 11, 15, 12) and analyzed using one-way ANOVA with Dunnett's post hoc test. (\**P* < 0.05, \*\*\*\**P* = 0.0001). **d** CRY2high, CRY2wt and CRY2low formed different amount of clusters on plasma membrane at different CRY2 concentrations. (CRY2high *n* = 48, CRY2wt *n* = 48, CRY2low *n* = 49) *Scale bars*, 5 μm

kinetics (half time 148 s) than CRY2(E490G), suggesting that the uncharged residues 490–498 may also aid CRY2 oligomerization. Though possessing very different tendencies for homo-oligomerization, all CRY2 derivatives with mutations at 490 and/or 489 retain CIB1-binding ability (Supplementary Fig. 5).

**C-terminal negative charges inhibit CRY2 homo-oligomerization**. After demonstrating that positive charges at CRY2 C-terminus could augment CRY2 oligomerization, we investigated whether negative charges would suppress CRY2 clustering. We designed a series of CRY2 mutants by adding 1–4 negatively charged residues to CRY2(Δ489–498), hereafter referred to as CRY2(1–488) (Fig. 3a). Increasing the number of negatively charged residues at the C-terminus of CRY2(1–488) yields more negative charges on the protein surface (Supplementary Fig. 2c). These variants were fused with mCh and transfected into COS7 cells. None of these cytosolic CRY2 mutants formed noticeable clusters upon blue light stimulation, similar to cytosolic CRY2wt (Supplementary Fig. 5). We have previously shown that CRY2 membrane recruitment via the CRY2–CIB1 interaction drastically amplifies CRY2 oligomerization[20]. In this work, we used this amplification system to detect any difference in homo-oligomerization of these mutants. Briefly, COS7 cells were co-transfected with CIB1-GFP-Sec61 and each mCh-tagged CRY2

derivative. Blue light was delivered at 2-s intervals. At *t* = 2 s, the cytosolic CRY2 was recruited to the ER membrane via CRY2-CIB1 hetero-dimerization, where it could aggregate into bright clusters (Fig. 3b and Supplementary Movie 4). At *t* = 20 s, CRY2wt almost completely aggregated into clusters and depleted diffusive CRY2 on the ER membrane, rendering the structure of the ER network indiscernible. By contrast, those mutants bearing C-terminal negative charges formed fewer clusters and the ER network remained visible. CRY2(1–488)EED and CRY2(1–488)EEDD, carrying 3 and 4 negative charges, respectively, yielded remarkably fewer and smaller clusters.

To quantify the differences in oligomerization among our panel of CRY2 variants, the cluster mass of each ER membrane-bound CRY2 was computed using Matlab and normalized against that of CRY2wt (Fig. 3c). CRY2 mutants that can form clusters in cytosol coalesced rapidly into large clusters after recruitment to ER membrane (Supplementary Fig. 6). Their cluster masses were similar and only slightly higher than that of CRY2wt, likely because membrane recruitment drastically amplified the clustering capacities for all these mutants. For the mutants with only negative charges at residues 489 and 490, the cluster masses were significantly lower than that of CRY2wt. CRY2(R489D) and CRY2(R489E) contained double negative charges at residues 489 and 490, and showed similarly low levels of clustering on ER membrane. Among the truncated mutants, the cluster masses

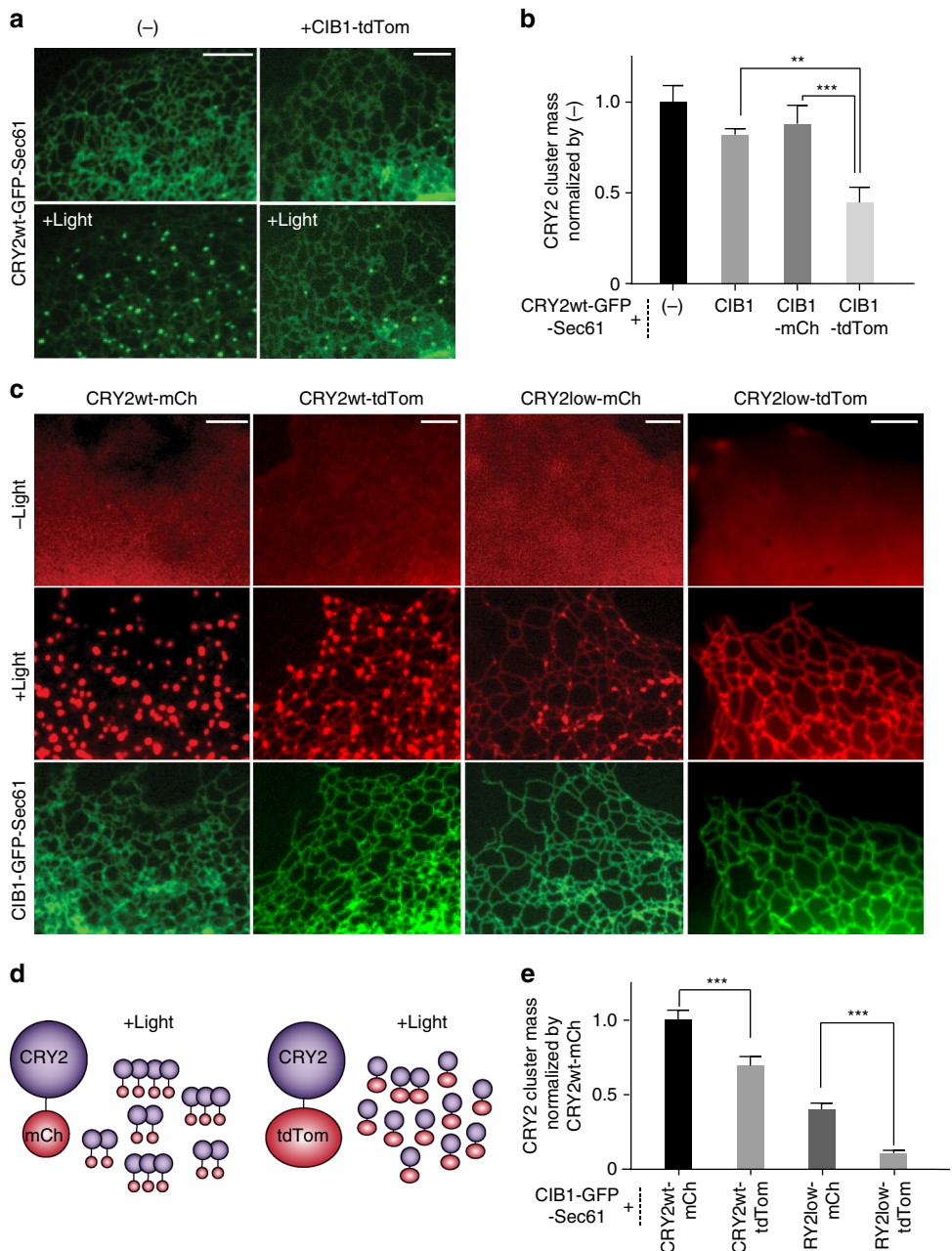

**Fig. 4** Fusion with a large fluorescent protein, tandem dimeric Tomato, further decreases CRY2low homo-oligomerization. Blue light was delivered at 5-s intervals for 100 s (**a**, **b**) or at 2-s intervals for 20 s (**c**, **d**). **a** In cells expressing both CRY2wt-GFP-Sec61 and CIB1-tdTom, blue light induced fewer clusters formation compared to cells expressing only CRY2wt-GFP-Sec61. **b** Quantification of the cluster mass of CRY2wt-GFP-Sec61, with or without co-expressing CIB1 fusion proteins, shows that CIB1-tdTom suppresses CRY2 cluster formation. ($n = 11$, 16, 11, 10). **c** After recruitment to the ER membrane via CRY2-CIB1 hetero-dimerization, CRY2wt-tdTom formed fewer clusters than CRY2wt-mCh, and CRY2low-tdTom did not form visible clusters on the ER network. **d** Illustration of CRY2 homo-oligomerization suppressed by the large protein tdTom. **e** The cluster mass of CRY2 on ER membrane after recruitment via CRY2–CIB1 hetero-dimerization was quantified and normalized against that of CRY2wt-mCh. ($n = 16$, 12, 15, 14). Results are presented as means ± s.e.m. and analyzed using one-way ANOVA with Dunnett's post *hoc* test. (**$P < 0.005$, ***$P < 0.0005$). *Scale bars*, 5 μm

decreased further with each additional negative charge (cluster mass of CRY2(1–488)E > CRY2(1–488)EE > CRY2(1–488)EED and plateaued at CRY2(1–488)EEDD). These findings strongly support that C-terminal negative charges can attenuate CRY2 aggregation, and this effect saturates with three negative charges at the CRY2 C-terminus.

By manipulating C-terminal charges, we developed CRY2 (E490R), referred to as CRY2high, which shows more rapid and extensive clustering than the previously reported clustering tool CRY2olig (CRY2(E490G)). In addition, we have developed a

CRY2 mutant with significantly suppressed clustering, CRY2 (1–488)EED, which we refer to as CRY2low.

To quantify the affinity of CRY2high, CRY2wt and CRY2low homo-oligomerization, we measured the concentration-dependent cluster formation on the plasma membrane. CRY2 mutants were directed to plasma membrane by tagging with CAAX, a cell membrane targeting sequence[47]. The two-dimensional protein concentrations are estimated *in situ* by relating fluorescence intensities in individual cells to absolute protein concentrations in western blot (detailed method in Supplementary

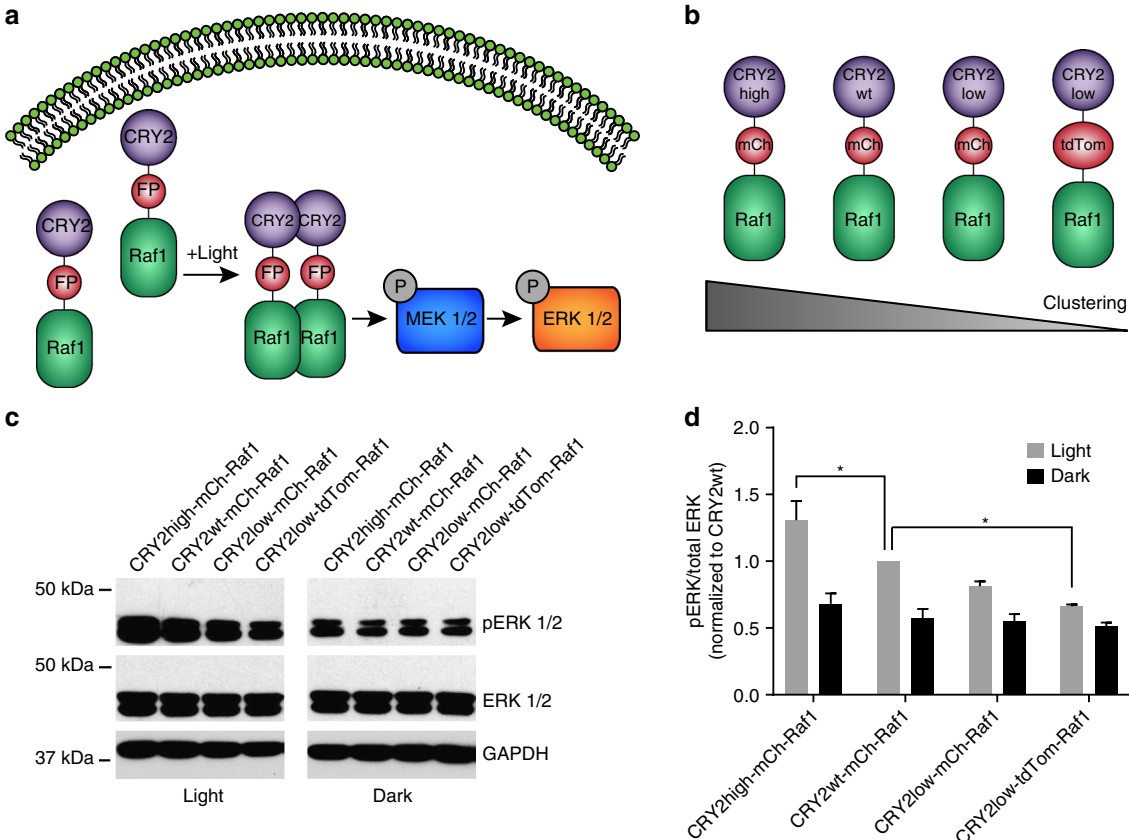

**Fig. 5** Optical control of Raf activation using CRY2 constructs with differential clustering capacities. **a** Illustrative scheme for light-induced activation of MAPK signaling pathway. **b** Light-activatable Raf1 with different CRY2 oligomerization propensities. **c** Cells expressing CRY2high-mCh-Raf1 yielded the highest level of pERK after light stimulation, while cells expression CRY2low-tdTom-Raf1 produced the lowest level of pERK after light illumination. Western blot analysis of phosphorylated ERK (pERK, Thr202 and Tyr204) was conducted after blue light illumination at 0.2 mW cm$^{-2}$ for 5 min. **d** The pERK value decreased from CRY2high-mCh-Raf1, CRY2wt-mCh-Raf1, CRY2low-mCh-Raf1 to CRY2low-tdTom-Raf1 after light stimulation. The pERK value in each transfection was averaged from two technical replicates from two independent experiments. The values were normalized against pERK from cells expressing CRY2wt-mCh-Raf1. Results are presented as means ± s.e.m. and analyzed using one-way ANOVA with Dunnett's post hoc test. (*$P < 0.05$)

Methods), with an average protein concentration $3.07 \times 10^4$ molecules μm$^{-2}$ correlated to average fluorescence intensity of 1900. With similar CRY2 concentrations, CRY2high-mCh-CAAX aggregated dramatically after one pulse of 100 ms blue light, while CRY2wt-mCh-CAAX clustered significantly less at the same condition, and CRY2low-mCh-CAAX only formed a few clusters (Supplementary Fig. 7a). The cluster masses were measured in ~ 50 cells for each mutant, and plotted against the CRY2 concentration. As shown in Fig. 3d, CRY2high, CRY2wt, and CRY2low have half-maximum concentrations of 4000, 13,000, and >30,000 μm$^{-2}$. As a control, we have shown that light-dependent clustering of all three mutants, CRY2high, CRY2wt and CRY2low, can be effectively inhibited by BIC1, the inhibitor that is recently identified to suppress CRY2 oligomerization[48] (Supplementary Fig. 7b).

**Fusion with tdTom further decreases CRY2low homo-oligomerization.** As minimal oligomerization of CRY2 is strongly favored to avoid complications during applications based on CRY2–CIB1 interactions, we sought to further suppress oligomerization of CRY2low. In our previous work, we demonstrated that bulky proteins fused with CIB1 suppressed CRY2 oligomerization through CRY2-CIB1 hetero-dimerization[20]. In order to take advantage of steric hindrance to further lower CRY2 oligomerization, we chose tandem dimeric Tomato (tdTom),

which at 60 kD is about twice the size of mCh. First, we examined whether tdTom can sterically hinder CRY2 clustering during CRY2–CIB1 dimerization. COS7 cells were transfected with CRY2wt-GFP-Sec61, which directly targets to the ER membrane, with or without CIB1-tdTom. Blue light was delivered at 5-s intervals for a total of 100 s. For cells expressing only CRY2wt-GFP-Sec61, blue light stimulation caused massive cluster formation on the ER membrane. However, cells expressing both CRY2wt-GFP-Sec61 and CIB1-tdTom showed fewer CRY2 clusters, and the ER network structures remained clearly visible (Fig. 4a). As control experiments, CIB1-mCh and CIB1 without any fusion tag were much less effective in suppressing the oligomerization of CRY2wt-GFP-Sec61 (Fig. 4b), which confirms the steric effect of tdTom. We also found that CIB1-tdTom inhibits CRY2 clustering in a concentration-dependent way. In cells with very low expression of CIB1-tdTom, CRY2 still formed many clusters on the ER membrane (Supplementary Fig. 8).

Next, we investigated whether tdTom directly fused to CRY2 was able to inhibit CRY2 oligomerization due to its steric hindrance (Fig. 4c, d). For this purpose, we co-transfected COS7 cells with CIB1-GFP-Sec61 and each of the following CRY2 constructs (CRY2wt-mCh, CRY2wt-tdTom, CRY2low-mCh, or CRY2low-tdTom) (Supplementary Movie 5). Cells were stimulated by blue light at 2-s intervals for 20 s. CRY2wt-mCh was recruited to the ER membrane and formed many large clusters; CRY2wt-tdTom formed slightly fewer clusters; CRY2low-mCh

formed even fewer clusters. The combination of CRY2low and tdTom dramatically reduced CRY2 clustering after recruitment to ER membrane, with very few or no clusters formed. The dense ER network was clearly visible, indicating that the CRY2–CIB1 interaction was not perturbed. Quantitative analysis of CRY2 cluster masses on ER confirmed that combining with tdTom can suppress CRY2 aggregation, with CRY2low-tdTom showing the least cluster formation (Fig. 4e). Therefore, CRY2low-tdTom exhibits very low oligomerization ability and would be ideal for applications that desire only CRY2–CIB1 hetero-dimerization.

**CRY2 variants tune the level of light-induced extracellular signal-regulated kinase (ERK) activation**. Next we tested whether CRY2 constructs of ranged oligomerization tendencies could be used to provide finer control of light-induced intracellular activity. The mitogen-activated protein kinase (MAPK), or ERK pathway is vital for cell proliferation, differentiation and apoptosis. The three-tiered protein kinase cascade RAF–MEK–ERK can be activated by the dimerization of Raf proteins, a family of serine/threonine protein kinase. Recent reports have demonstrated optical control of Raf1 by fusing Raf1 with CRY2wt[31, 32], where light-induced oligomerization of CRY2 activated Raf1 and then phosphorylated MEK and ERK (Fig. 5a). Here, we fused Raf1 to four different CRY2 constructs with a broad range of clustering capacities, CRY2high-mCh, CRY2wt-mCh, CRY2low-mCh and CRY2low-tdTom (Fig. 5b). We hypothesized that CRY2 constructs with stronger oligomerization could induce more Raf1 activation at identical conditions.

To test our hypothesis, each of the four CRY2-Raf1 constructs was transfected into HEK293T cells and the extent of light-induced Raf1 activation was evaluated by western blot analysis of phosphorylated ERK (pERK, Thr202 and Tyr204). After 6 h serum starvation, HEK293T cells were illuminated with blue light at $0.2 \ mW \ cm^{-2}$ for 5 min, followed by immediate cell lysis and western blot analysis. As a control, pERK levels were probed in transfected cells that were kept in dark. As shown in Fig. 5c, the amount of pERK was highest in cells expressing CRY2high-mCh-Raf1, and decreased from cells transfected with CRY2wt-mCh-Raf1, CRY2low-mCh-Raf1 to CRY2low-tdTom-Raf1. Compared with the dark control, cells subjected to light stimulation showed greater ERK phosphorylation. The amount of phosphorylated ERK in each sample was measured and averaged from two technical replicates from two independent experiments. The values were normalized to the sample expressing CRY2wt-mCh-Raf1 (Fig. 5d). Indeed, quantitative analysis confirmed that CRY2high-mCh-Raf1 activated Raf1 more efficiently than CRY2wt-mCh-Raf1, yielding stronger phosphorylation of ERK. Conversely, CRY2low-tdTom-Raf1 only showed a slight increase in pERK levels compared to the corresponding dark control, confirming that CRY2low-tdTom, with the least clustering ability, barely activated MEK-ERK signaling. Our results demonstrate that by using CRY2 constructs with differential clustering capacities, optically controlled events can be tuned to a desired level.

## Discussion

Here we have shown that electrostatic charges at the N- and C-termini drastically affect CRY2 interactions with CIB1 or CRY2, respectively. We found that positive charges at the N-terminus of CRY2 are required for light-mediated CRY2-CIB1 dimerization. On the other hand, C-terminal positive charges promote CRY2 homo-oligomerization and negative charges suppress it. These findings indicate that CRY2-CIB1 and CRY2-CRY2 interactions can be modulated by engineering two distinct interfaces of CRY2.

The mechanisms responsible for CRY2 oligomerization appear to be quite complex. We have shown that CRY2 oligomerization could be reduced by C-terminal negative charges. However, multiple negative charges at the C-terminus did not completely abrogate CRY2–CRY2 interaction, which indicates that there could be additional sites or other types of forces involved in CRY2 oligomerization. In addition, we have observed that large clusters formed by CRY2(E490G), CRY2high, CRY2(E490H) and CRY2 (R489E,E490R) persisted for at least 1 h in dark (Supplementary Fig. 9), while smaller aggregates dissociated within 10 min. This observation suggests that large clusters may be stabilized by interactions at surfaces that are not light-sensitive, but occur secondarily, after microscopic oligomerization formation, due to avidity affects[49]. More investigation is needed to uncover the complex mechanisms of CRY2 oligomerization.

In this report, we presented several CRY2 constructs with a wide range of oligomerization propensities, with CRY2low-tdTom displaying minimal oligomerization and CRY2high showing drastically enhanced aggregation capacity. The formation of CRY2 clusters may interfere with desired protein regulation via CRY2–CIB1 hetero-dimerization. For instance, Nihongaki et al. failed to engineer an optically reconstituted Cas9 using CRY2–CIB1 hetero-dimerization[47]. They hypothesized that CRY2 oligomerization interfered with the interaction between split Cas9 fragments. Therefore, the formation of CRY2 aggregates during CRY2–CIB1 interaction adds another layer of complexity and induces undesired effects. We propose that these undesired effects can be mitigated by using CRY2low-tdTom. As shown in Fig. 4e, CRY2low-tdTom did not form any visible puncta after recruitment to the ER membrane via CIB1 interaction. Therefore, we expect that CRY2low-tdTom will be generally useful in cases where CRY2 clustering would prove detrimental.

CRY2high, which shows enhanced clustering, also can be a useful optogenetic tool to optically control signaling pathways that can be activated by oligomerization of a single component. Several recent reports have shown that Raf1 fused with wild-type CRY2 enables optical activation of MAPK signaling. We compared the efficiency of CRY2high-mCh-Raf1 and CRY2wt-mCh-Raf1 and found that CRY2high-mCh-Raf1 induces stronger phosphorylation of ERK than CRY2wt-mCh-Raf1 (Fig. 5). Therefore, CRY2high would be useful for optical control of signaling proteins which can be activated by dimerization/oligomerization.

In summary, we find that two light-induced behaviors of CRY2, hetero-dimerization with CIB1 and homo-oligomerization, map to two different regions on the protein. We also characterize mutants that specifically promote or suppress homo-oligomerization. Our findings contribute to the understanding of the molecular mechanisms underlying how the plant photoreceptor CRY2 responds to light, and will also help guide efforts on further engineering of CRY2 for optical control of biological systems.

## Methods

**Plasmids**. In this work, we employ the PHR of CRY2, consisting of amino acids 1–498, and a truncated version of CIB1 (a.a 1–170), both gifts from Professor Chandra Tucker at the University of Colorado Denver. The plasmid containing the ER-targeting Sec61 was a gift from Professor Jonathan Weissman from the University of California, San Francisco. CRYwt-mCh-Raf1 was made in our previous work[20]. All other plasmids were generated using DNA ligation or the InFusion cloning kit (Clontech, Mountain View, CA). Descriptions of plasmid construction in this work are given in Supplementary Table 1.

**COS7 cell culture and transfection**. COS7 monkey fibroblast cells (obtained from ATCC) were cultured on a PLL-coated glass coverslip and maintained in Dulbecco's modified Eagle's medium (Gibco) supplemented with 10% fetal bovine serum (FBS, Atlanta Biologicals) and 1% penicillin/streptomycin (Gibco). Cells were maintained in a humidified atmosphere containing 5% $CO_2$ at 37 °C. 24 h

prior to imaging, at 70–90% confluency, the cultures were transiently transfected using Lipofectamine 2000 (Thermo Fisher) following the manufacturer's protocol.

**HEK293T cell culture and transfection**. HEK293T cells (obtained from ATCC) were cultured on a 12-well plate in Dulbecco's modified Eagle's medium/Nutrient Mixture F-12 (DMEM/F-12, Gibco) supplemented with 10% fetal bovine serum and 1% penicillin/streptomycin (Gibco). Cells were maintained in a humidified atmosphere containing 5% $CO_2$ at 37 °C. 24 h prior to light activation, cells were transiently transfected using Lipofectamine 2000 (Thermo Fisher) following the manufacturer's protocol.

**Live-cell imaging**. Live-cell imaging was carried out using an epi-fluorescence microscope (Leica DM16000B) equipped with adaptive focus control and an on-stage incubator chamber (Tokai Hit GM-8000), which maintained 5% $CO_2$, humidity, and a temperature of 37 °C for the duration of the imaging experiment. Images were acquired using a 100 × oil-immersion objective (Leica, HCX PL APL, n.a. 1.4) and a light-emitting diode (LED) light source (Lumencor Sola, Beaverton, OR). CRY2 activation was triggered either by a single 1 s blue light exposure or a series of 200 ms pulses delivered at 2- or 5-s intervals, at 9.7 W cm$^{-2}$. GFP fluorescence signal was detected using a commercial GFP filter cube (Leica, excitation 472/30, dichroic mirror 495, emission 520/35). mCh was excited using green light (~550 nm, 2.5 W cm$^{-2}$), and its fluorescence signal was detected using a commercial Texas Red filter cube (Leica, excitation 560/40, dichroic mirror 595, emission 645/75).

**Image processing**. The expression level of CRY2 was measured in each cell in the first image frame ($t = 0$, prior to blue light illumination) using ImageJ. The expression for each cell was calculated as the average intensity of CRY2 minus the image background $I_{background}$. $I_{background}$ is defined as the average intensity of an image from blank areas (outside cells).

Cluster quantification in the cytosol or on the ER membrane was performed using a custom-written Matlab program, which has been described in our previous work[18]. The Matlab program removes the image noise to enhance the detection of clusters by performing Gaussian smoothing. The background is calculated using 10 × 10 pixel mean filter and subtracted from the original image. The Matlab algorithm automatically detects the locations of individual CRY2 clusters by picking the local maxima. The intensity of each cluster is calculated by summing the intensity values of all the pixels in a 7 × 7 pixel region surrounding the center pixel. For each image frame, the cluster mass is calculated as the sum of all cluster intensities in the cell, normalized by the cell size and average cell intensity. The average cluster mass for each CRY2 derivative was then normalized by that of CRY2wt at identical imaging/stimulation conditions.

**Generating protein models and surface charge maps**. Comparative modeling for protein structure was conducted for CRY2wt and all CRY2 derivatives investigated in this work. The template structure is the PHR domain of *Arabidopsis* CRY1 (PDB code 1U3C:A) with ~ 60% sequence identity to CRY2, ensuring the accuracy of the modeling. For each CRY2 mutant, the sequence alignment was performed using Align2D. For each template-target pair, a total of 100 models were generated with random initial condition followed by simulated annealing algorithm with Modeller[40]. The 100 candidate models were subsequently assessed by the DOPE score[41]. For each CRY2 mutant, the model with the lowest DOPE score is used for final visualization and electrostatic analysis. The distribution of electrostatic potential on the surface of each CRY2 mutant was calculated based on the Poisson–Boltzmann equation using Swiss-pdb Viewer Version 4.1.0. Positive charge is shown in blue and negative charge in red.

**Immunoblotting**. HEK293T cells were transfected as described above and allowed to recover overnight in culture medium. After recovery, cells were serum starved for 6 h to minimize background kinase activity and then exposed to blue light using a homemade LED array for 5 min at 0.2 mW cm$^{-2}$; control cells were kept in dark. Cultures were then moved to ice and immediately lysed with RIPA buffer supplemented with protease and phosphatase inhibitors. Protein concentrations were determined using a Bradford assay with a bovine serum albumin standard curve. Electrophoresis samples were prepared in 4x Laemmli buffer (Bio-Rad) with β-mercaptoethanol and boiled at 95 °C for 10 min, then subjected to SDS-PAGE on a 10% acrylamide gel. Protein was transferred to a polyvinylidene fluoride membrane using the Trans-Blot Turbo system (Bio-Rad), and then probed using anti-ERK or anti-pERK antibody (1:5000 dilution, 9102 and 4370, Cell Signaling Technology). HRP-conjugated secondary antibodies were used for protein band detection (1:10,000 dilution, 7074, Cell Signaling Technology). Anti-GAPDH (1:5000 dilution, 2118, Cell Signaling Technology) was used as a loading control as well as to normalize across preparations. Blots were quantified by densitometry using ImageJ. Full scans of blots are presented in Supplementary Fig. 10.

**CRY2 concentration measurement**. The mCh fluorescence intensity was first correlated to absolute CRY2 concentrations by measuring and quantifying the average mCh intensity and concentration of CRY2 in COS7 cells expressing Flag-CRY2low-mCh-CAAX. The concentration of CRY2 was calculated by (the total protein amount of CRY2)/(the sum of cell areas). The average mCh intensity in the cell culture was obtained by measuring the fluorescence intensity of 630 cells randomly taken across the culture using 100x objective. The sum of cell areas in one well of 12-well plate was measured and quantified from 20 mCh images randomly taken throughout the culture using 20x objective. The total amount of CRY2 protein in one well of 12-well was obtained by calculating the mCh amount using various amount of purified mCh proteins as the calibration in a western blot analysis. Please see Supplementary Fig. 11 and Supplementary Methods for experiment and quantification details.

**Data availability**. All the data generated or analyzed during this study are included in this published article and in Supplementary Information files. They are available from the corresponding author upon reasonable request.

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

## Acknowledgements

We thank Dr. Chandra Tucker (University of Colorado Denver) for providing CIB1-GFP-Caax and CRY2-mCherry; we thank Dr. Jonathan Weissman (University of California, San Francisco) for providing the Sec61 plasmid. We thank Dr. Chentao Lin (University of California, Los Angeles) for providing GFP-BIC1 construct. We also thank Zhuoluo Feng (Currently in China Resources) for assistance in constructing the controllable blue LED array. We thank Xin Zhou (Stanford University) for comments on the manuscript. This work was supported by the US National Institute of Health (DP2-NS082125) and a Packard fellowship in Science and Engineering.

## Author contributions

L.D., M.Z.L., B.C.: Conceived and designed the experiments. L.D., J.H., C.M.: Made the plasmids. L.D., J.H., Q.O., H.-Y.L., N.K., V.A.: Performed the experiments. L.D., J.H. analyzed the data and wrote the "Methods" section. L.D., M.Z.L., B.C.: Wrote the manuscript. J.H., Q.O., H.-Y.L., C.M.: Discussed the results and commented on the manuscript.

## Additional information

**Competing interests:** The authors declare no competing financial interests.

