## [Peer Review File · Nature Communications]

Reviewers' comments:

Reviewer #1, an expert in light regulated proteins (Remarks to the Author):

Duan and co-workers study the oligomerization and heterodimerization behavior of a plant cryptochrome (specifically, cryptochrome 2 of *Arabidopsis*, with a focus on its PHR domain, here simply referred to as "CRY2") in a heterologous mammalian expression systems. This domain has attracted fame in the past years as a clustering and recruitment optogenetic tool; however, molecular mechanisms, e.g. interaction sites or interfaces, underlying the association behaviors used in optogenetics have remained poorly understood. The authors address this knowledge gap and show that heterodimerization with CIB can be pinpointed to the N-terminus of the PHR domain, whereas the C-terminus is critically implicated in oligomerization. The authors then identify in great detail the electrostatic determinants and their functional consequences, including an application that is the regulation of the MAP/ERK pathway.

This is an overall excellent work that is systematic, timely, and well written. I support publication "as is" and have only a few minor suggestions and questions for the authors:

- The authors take great care in referencing many important studies in the field; however, they appear to focus mainly if not exclusively on work that describes heterointeractions of LOV domains, phytochromes etc. In the context of a study on the homointeraction of CRY2 it may make sense to also (or even preferably) mention homomeric LOV and phytochromes interactions, also because were established in the context of the MAPK/ERK pathway.
- Page 6, line 112: The authors state an intensity of 10 W/cm² in the clustering experiments; this value is rather high compared to values used for functional data in the second half of the manuscript (0.2 mW/cm²); what was the reason for choosing this intensity?
- Page 7 and following: The authors present a very detailed analysis of the impact of protein truncation on electrostatic surface properties at the C-terminus using modelling. My questions are: How was the surface area chosen that is evaluated here (e.g. boxes in Figure 1h?). What mechanisms are at play that permit the single glutamate 490 residue to alter the surface potential "by suppressing [of both]" the neighboring, much larger arginines (e.g. deprotonation? orientation away from the surface? isn't glu490 only critical in suppressing the charge of R489 as the surface becomes negative again in delta489-498?). How can one know that the modelling methodology applied is of sufficient accuracy for obtaining realistic conformers and charges at the surface (e.g. are solvents and molecular dynamics applied, if not, then why not?)?
- Page 7, line 152: I believe this sentence is a duplication.
- Page 8, line 161 and 164: Is Arg-489 indeed required for oligomerization or does it enhance it?
- Page 10, line 229 and following: The authors state that addition of negative charges gradually impacts the oligomerization behavior with "a plateau". This statement hinges on the four rightmost bars of Figure 3c; notably E489 performs weaker than D489 leaving the impression of a trend and all values are very close together. Can this influential statement be further supported by meaningful statistical analysis and how can the functional similarity of D489 and E489E490 etc. be explained?
- Page 2 of supplement, Figure S1: I must have missed it: How is I(background) defined?
- Probably I am wrong but the figure(s) (panels) are not called in sequence (e.g. 1e before 1d, and S4 before S3...)

Reviewer #2, and expert in CRY2/CIB interactions (Remarks to the Author):

The ms by Duan described a study for how to improve efficiency or photo-responsiveness of light-dependent CRY2 oligomerization and CRY2-CIB1 interaction, two tools very widely used in optogenetics. These authors used fluorescence protein-based protein clustering assay as the readout to assess the structural elements underlying these two types of protein-protein interactions. They found that positive charges at both ends of the PHR domain of CRY2 have generally positive effects on the protein-protein interaction, whereas negative charges at the two ends of the PHR domain of CRY2 usually inhibit clustering. They also used site-specific mutants to verify these discoveries. Resulting from these analyses, the authors obtained two new versions of CRY2-PHR fragment that appear useful to the future optogenetics studies, the CRY2^{High} and CRY2^{low}. And they used those new constructs in a proof-of-principle experiment to successfully demonstrate that the new CRY2 constructs can be used to study RAF dimerization-dependent RAF-MEK-ERK signaling. The discovery provides new insights to the photodimerization of plant CRY and provide new optogenetics tools for biomedical researches. It should be interesting to the broad readership of this journal.

Specific comments:

1. All quantification of CRY2 dimerization or oligomerization was obtained by digitizing the microscopic "dots", which is generally considered less quantitative than that obtained by other types of semi-quantitative methods. The authors may want to present an independent line of evidence for the effects of CRY2^{High} and CRY2^{low}. For example, a semi-quantitative co-IP experiment may provide at least a different view of "how much" better or worse of the photoreaction of those constructs in comparison to the wild-type CRY2 constructs. They may present a very different result, but it may not affect the authors conclusion as long as the trend is the same.
2. It has been recently reported that a CRY2 inhibitor BIC1 can suppress CRY2 photodimerization (Science 354:343-347). The authors may want to test how BIC1 affect light responsiveness of CRY2^{High} and CRY2^{low} in at least a control experiment.

Reviewer #3, an expert in light-regulated protein engineering (Remarks to the Author):

The light-promoted heterodimerization of the *Arabidopsis thaliana* photoreceptor cryptochrome 2 (Cry2) with its interacting partner CIB1 underpins many optogenetic applications that rely on light-induced protein-protein interactions. In addition to forming Cry2:CIB1 heterodimers, Cry2 also shows propensity to form higher-order oligomers upon light exposure. For certain applications, this latter activity is desirable but for others it disturbs. An earlier report by Tucker and colleagues achieved a significant step towards understanding and hopefully untangling the concurrent hetero- and homo-association reactions: the mutation E490G, denoted Cry2^{olig}, in the very C terminus of the Cry2 construct routinely used greatly increased clustering capability of Cry2. Cui, Lin and coworkers now follow up and present a suite of Cry2 derivatives that show more or less clustering proficiency while retaining the ability to interact with CIB1 in light-activated manner. This palette of light-gated actuators promises to be useful where exclusively one or the other of the two light-driven responses of Cry2 is called for. The authors construct several N- and C-terminal truncations/substitutions of Cry2, thus altering charge distribution in these regions of the protein; to test homo-/hetero-association, CIB1 is directed to the ER membrane and Cry2 associates in light-dependent manner. Clustering is assessed by puncta formation of fluorescently labeled Cry2. One variant, E490R, shows similar clustering ability as Cry2^{olig} (i.e. E490G), whereas placement

of several negative charges in the C terminus reduced clustering. When applied to Raf signaling, the different variants activated the pathway to different extent. Altogether this is an interesting, largely well written and potentially quite useful contribution. However, there are a couple of aspects that need to be addressed prior to publication. It appears that N-terminal modifications of Cry2 had a marginal effect, especially when compared to the C-terminal modifications. I hence feel that the N-terminal variants could be given short shrift and panels 1b, d, e could be safely removed (or, relegated to Suppl. Mat.) without significant loss to the quality of the manuscript. In a related vein, I suggest toning down on the use of the structural models, both in the figures and the text. Whereas the initial homology model and its surface charge representation have some merit, I am not convinced by the derivative models that follow (esp. in fig. 1h). The authors just chop off residues in silico but assume that the overall structure of Cry2 is left unaltered; this is much too simplistic an assumption, and I recommend removing altogether these structural models. Moreover, the message the authors want to convey, namely that they add, remove or invert charges, can easily be grasped without these models (except for the original one which is fine). Furthermore, I note that throughout the manuscript the Cry2:Cry2 and the Cry2:CIB1 interactions have only been studied in cellular settings which makes it hard to derive precise quantitative conclusions about the efficiency of the various Cry2 species for either reaction. Specifically, differences in intracellular expression, fraction of active protein and trafficking will all factor into the effective intracellular concentration; as a consequence, apparent differences between variants could be solely due to these aspects but need not necessarily reflect a genuine difference in association constants (this issue is nicely discussed in PMID 26137467). Even if real, it would be informative to know in terms of association constants by which factor the variants differ. It is at the editor's discretion to decide on this point but the complete absence of detailed studies with purified Cry2 proteins is a weak point of the manuscript. Despite that, the proof-of-principle experiments indicate that the new Cry2 variants could be quite useful at the practical level (for whatever underlying mechanistic reason); this is particularly true for the Cry2 variants with reduced clustering proficiency. By contrast, the new variant E490R with more pronounced clustering is only marginally better than the existing Cry2olig. In summary, I believe this manuscript could become suitable for publication in 'Nature Communications' after taking care of these major and the below minor points.

34/35: does not make much sense to mention 'the light-oxygen-voltage (LOV) domain', only to then go on and mention Vivid which also happens to be a LOV protein; presumably the authors mean the paradigmatic *A. sativa* phot1 LOV2 domain and should modify their text accordingly

54: mutation underlying Cry2olig should be explicitly named on first occasion

69: explicitly state that residues 489-490 are at the C term

p6: as discussed above, the apparent differences b/w variants in their interaction with CIB1 might be due to expression etc.-- has this been checked? if so, how? Even so, I don't find the effect all too dramatic and hence deem this section a weaker part of the paper.

113: panel e of fig. 1 is cited before panel d; the same is true for panels g and h further down

123-130: not sure whether the chosen setup is suited to address relative clustering efficiency; rather, to do so, one would need to go to a setting where Cry2 wt is on the verge of clustering to tease out differences of the variants, if any. Could one go back to the setting of the original Bugaj paper where they first reported Cry2 clustering?

p7.143: throughout the paper, please refrain from calling the structural models 'structures' -- for what they are worth, they are models and hence should be called thus

172: depending on pKa of residue H490, it may be charged or not, so the statement '... two positive charges ...' could potentially be inaccurate. Also, why was lysine (E490K) not tested as well? (seems straight-forward)

p9 middle para: it would be informative to include actual numbers when comparing the variants, rather than saying 'greater cluster mass' or 'similar clustering capabilities' etc.

213-218: cf. above, how do we know whether these differences are indeed due to differences in association constants as opposed to conc. differences of active Cry2? at the very least, this should be discussed

220: reference to Matlab not needed, should be explained in Methods

235: again, what about E490K?

287/292: references should presumably be to fig. 5

334: fig. 6??

534: `... are removed ...`

542: `... amino acid ...`

545: `... residues 490 to 487 ...`; does not make much sense, does it?

Response to reviewers' comments:

We thank the reviewers for their constructive comments that have helped us to significantly improve the manuscript. We have performed new experiments to address reviewer's concerns. Below, we address the points raised by each reviewer.

Reviewer 1

Duan and co-workers study the oligomerization and heterodimerization behavior of a plant cryptochrome (specifically, cryptochrome 2 of Arabidopsis, with a focus on its PHR domain, here simply referred to as "CRY2") in a heterologous mammalian expression systems. This domain has attracted fame in the past years as a clustering and recruitment optogenetic tool; however, molecular mechanisms, e.g. interaction sites or interfaces, underlying the association behaviors used in optogenetics have remained poorly understood. The authors address this knowledge gap and show that heterodimerization with CIB can be pinpointed to the N-terminus of the PHR domain, whereas the C-terminus is critically implicated in oligomerization. The authors then identify in great detail the electrostatic determinants and their functional consequences, including an application that is the regulation of the MAP/ERK pathway.

This is an overall excellent work that is systematic, timely, and well written. I support publication "as is" and have only a few minor suggestions and questions for the authors:

--- We very much appreciate Reviewer 1's comments on the significance and support of publication of our work.

The authors take great care in referencing many important studies in the field; however, they appear to focus mainly if not exclusively on work that describes heterointeractions of LOV domains, phytochromes etc. In the context of a study on the homointeraction of CRY2 it may make sense to also (or even preferably) mention homomeric LOV and phytochromes interactions, also because were established in the context of the MAPK/ERK pathway.

--- We agree with the reviewer's comments. In the revised manuscript, we included previous reports using homo-dimerization of LOV domain and phytochrome. We also modified the corresponding text as "Optical approaches use photosensory domains that undergo light-induced protein-protein hetero-dimerization, such as the LOV (light-oxygen-voltage) domain of *Avena sativa* phototropin 1, *Arabidopsis* phytochrome B and *Arabidopsis* cryptochrome 2, or protein-protein homo-interaction, such as the LOV domain of *Vaucheria frigida* aureochrome1, *Synechocystis* phytochrome 1 and *Arabidopsis* cryptochrome 2."

Page 6, line 112: The authors state an intensity of 10 W/cm² in the clustering experiments; this value is rather high compared to values used for functional data in the second half of the manuscript (0.2 mW/cm²); what was the reason for choosing this intensity?

--- The high intensity 9.7 W/cm² is a typical intensity for a standard fluorescent microscope equipped with a 60x or 100x objective (generally about 1-100 W/cm² depending on the filters used¹). The high light intensity was used intermittently (200 ms per 2s interval, 200 ms per 5s interval or a single pulse). Though a low level of blue light (0.2 mW/cm²) by a custom-built LED array is sufficient to stimulate CRY2 activities as shown previously^{2, 3}, fluorescence imaging using blue light at 9.7 W/cm² is necessary to visualize GFP signal in our experimental setup.

Page 7 and following: The authors present a very detailed analysis of the impact of protein truncation on electrostatic surface properties at the C-terminus using modelling. My questions are: How was the surface area chosen that is evaluated here (e.g. boxes in Figure 1h?). What mechanisms are at play that permit the single glutamate 490 residue to alter the surface potential "by suppressing [of both]" the neighboring, much larger arginines (e.g. deprotonation? orientation away from the surface? isn't glu490 only critical in suppressing the charge of R489 as the surface becomes negative again in delta489-498?). How can one know that the modelling methodology applied is of sufficient accuracy for obtaining realistic conformers and charges at the surface (e.g. are solvents and molecular dynamics applied, if not, then why not?)?

--- To evaluate the charge effect at CRY2 C-terminus, the surface area chosen includes all of R487, R489 and E490. As shown in the figure below, Fig.R1a is the wireframe diagram of CRY2wt. The C-terminal tail of CRY2wt(a.a.486-498), is marked in orange in the ribbon diagram of CRY2wt (Fig.R1b) and corresponds to the indicated area in surface potential diagram (Fig.R1c). Fig.R1d is the zoomed-in image of C-terminus surface potential map with rotated views from two sides. In the final presentation of surface charge (Fig.R1e), the surface was set semi-transparent so that charges from both sides can be visualized.

Figure R1 | Modeling of CRY2wt structure and surface charge map. (a) Wireframe diagram of CRY2wt. (b) Ribbon diagram of CRY2wt. (c) Surface potential diagram of CRY2wt. (d) The zoomed-in image of CRY2wt C-terminus surface potential with rotated views from both sides. (e) The final presentation of surface charge.

The original statement “the negative charge at Glu-490 is critical in suppressing the

positive charges at Arg-489 and Arg-487” is misleading. We mean that E490 is critical in balancing the charges in the local surface area. The deletion of E490 leaves the area with purely positively charged amino acids. Indeed, as the reviewer pointed out, Glu-490 is mainly critical in suppressing the charge of R489. As the surface becomes negative again in CRY2(Δ 489-498), the CRY2 oligomerization tendency is low. We have changed the text as “These models suggest that, in this region of the molecule (shown in the insets), the negative charge at Glu-490 is critical in counterbalancing the positive charge at Arg-489 in enhancing CRY2 oligomerization.”

Regarding the CRY2 modeling, CRY2 has a very high sequence identity (~60%) to CRY1, which is the template structure for homology modeling. Previous studies show that homology structure models based on more than 50% sequence identity are generally considered high-accuracy models. The qualities of high-accuracy homology models are comparable to medium-resolution NMR or low-resolution X-ray structures, which have ~1Å RMSD for the main-chain atoms⁴. The surface charge pattern of the CRY2 is primarily determined by the positions of polar amino acid residues in the structure, which is determined by the main-chain conformation. The solvent effect and molecular dynamics are included in the model building. Solvent effect is included in two ways. (1) The electrostatic surface in this study is obtained using an implicit solvent model, Poisson-Boltzmann; and (2) the CRY2 homology models are assessed by a high-accuracy statistical potential DOPE, which includes the solvent effects from the statistics of experimental structures. The molecular dynamics is an integral part of MODELLER's model building protocol, where the solvent effects are included in the form of homology derived restraints and statistical potentials.

Page 7, line 152: I believe this sentence is a duplication.

--- In the experiment to test how CRY2 C-terminal truncations affect light-induced homo-oligomerization, COS7 cells were **singly**-transfected with each of mCh-tagged CRY2 variants. While in the previous experiment described in the same paragraph, cells were **co**-transfected with CRY2 variants and CIB1-GFP-Sec61. To better clarify the difference in these two experiments, we changed the corresponding text as “COS7 cells were **co**-transfected with mCh-tagged truncated CRY2 variants along with CIB1-GFP-

Sec61 (Supplementary Fig. 3).” “Next, to characterize the effects of CRY2 C-terminal truncations on light-induced homo-oligomerization, COS7 cells were **singly-transfected** with mCh-tagged CRY2wt, CRY2(E490G) (CRY2olig), or each truncated CRY2 mutant respectively.”

Page 8, line 161 and 164: Is Arg-489 indeed required for oligomerization or does it enhance it?

--- We thank the reviewer for pointing out our inaccurate usage of “required”. Arg-489 significantly enhances oligomerization but is not absolutely required. We have tried further truncated mutations CRY2(Δ 488-498) and CRY2(Δ 487-498). Although neither CRY2(Δ 488-498) or CRY2(Δ 487-498) forms clusters in cytosol, they can still form some clusters after being recruited to the ER membrane. Therefore, we change the text as “Additional removal of the adjacent residue, the positively charged Arg-489, in CRY2(Δ 489-498) then abolished the cytosolic aggregation, indicating that Arg-489 can **greatly enhance** CRY2 aggregation.”

Page 10, line 229 and following: The authors state that addition of negative charges gradually impacts the oligomerization behavior with “a plateau”. This statement hinges on the four rightmost bars of Figure 3c; notably E489 performs weaker than D489 leaving the impression of a trend and all values are very close together. Can this influential statement be further supported by meaningful statistical analysis and how can the functional similarity of D489 and E489E490 etc. be explained?

--- We performed one-way ANOVA with Dunnett’s post hoc test between any two of CRY2(1-488)E, CRY2(1-488)EE, CRY2(1-488)EED and CRY2(1-488)EEDD (number of cells n=13,11,15,12) and the results are shown below (Fig. R2). Indeed, CRY2(1-488)EED and CRY2(1-488)EEDD formed significantly fewer clusters than CRY2(1-488)E and CRY2(1-488)EE, while there was no significant difference between CRY2(1-488)EED and CRY2(1-488)EEDD (*P<0.05, **P<0.005). The analysis confirms the trend that the cluster masses decreased with each additional negative charge and plateaued at CRY2(1-488)EEDD. T-test was performed to compare CRY2(R489D) and CRY2(R489E) and showed no significant difference (p=0.59). Furthermore, mutants

with two C-terminal negative charges, CRY2(R489D) and CRY2(1-488)EE showed similar cluster masses. Therefore, negative charge, whether it is from aspartic acid or glutamic acid, significantly suppresses CRY2 homo-interactions.

Figure R2 | one-way ANOVA with Dunnett's post hoc test between any two of (1-488)E, (1-488)EE, (1-488)EED and (1-488)EEDD (n=13,11,15,12) (*P<0.05, **P<0.005).

Page 2 of supplement, Figure S1: I must have missed it: How is I(background) defined?

--- $I_{\text{background}}$ is defined as the average intensity of an image from blank areas (outside cells). We added more description in the method Image Processing section.

Probably I am wrong but the figure(s) (panels) are not called in sequence (e.g. 1e before 1d, and S4 before S3...)

--- We thank the reviewer for pointing this out. We have adjusted the figures so that the figures and panels are cited in sequence.

Reviewer 2

The ms by Duan described a study for how to improve efficiency or photo-responsiveness of light-dependent CRY2 oligomerization and CRY2-CIB1 interaction, two tools very widely used in optogenetics. These authors used fluorescence protein-based protein clustering assay as the readout to assess the structural elements underlying these two types of protein-protein interactions. They found that positive charges at both ends of the PHR domain of CRY2 have generally positive effects on the protein-protein interaction, whereas negative charges at the two ends of the PHR domain of CRY2 usually inhibit clustering. They also used site-specific mutants to verify these discoveries. Resulting from these analyses, the authors obtained two new versions of CRY2-PHR fragment that appear useful to the future optogenetics studies, the CRY2^{High} and CRY2^{low}. And they used those new constructs in a proof-of-principle experiment to successfully demonstrate that the new CRY2 constructs can be used to study RAF dimerization-dependent RAF-MEK-ERK signaling. The discovery provides new insights to the photodimerization of plant CRY and provide new optogenetics tools for biomedical researches. It should be interesting to the broad readership of this journal.

--- We thank the reviewer for his/her positive review of our work.

Specific comments:

1. All quantification of CRY2 dimerization or oligomerization was obtained by digitizing the microscopic “dots”, which is generally considered less quantitative than that obtained by other types of semi-quantitative methods. The authors may want to present an independent line of evidence for the effects of CRY2^{High} and CRY2^{low}. For example, a semi-quantitative co-IP experiment may provide at least a different view of “how much” better or worse of the photoreaction of those constructs in comparison to the wild-type CRY2 constructs. They may present a very different result, but it may not affect the authors conclusion as long as the trend is the same.

--- We thank the reviewer for the suggestion. In the last two months, we have conducted co-IP experiments in order to probe the light-dependent homo-interaction of CRY2^{high}, CRY2^{wt} and CRY2^{low}. The homo-association of full length CRY2 has been beautifully presented using co-IP experiments in a recent work⁵. We are grateful to Prof. Chentao Lin, who has provided us with detailed co-IP protocols and conditions for the co-IP experiment. Nevertheless, the co-IP experiment for the PHR domain of CRY2 has not yet been reported, so we set out by optimizing the experimental conditions, including transfection condition, light illumination duration and cell lysis handling in light or dark. Using a strategy similar to the previous report⁵, we constructed Flag-CRY2-mCh and Myc-CRY2-mCh. Both plasmids were co-expressed in HEK293T cells and Flag antibody-bound beads were used to precipitate Flag-CRY2-mCh, and the associated Myc-CRY2-mCh was probed with Myc antibody by the co-IP assay (see detailed procedure in **Materials and Methods** paragraph). All the CRY2 constructs we used in the following experiments were PHR domain.

Optimizing the transfection conditions

The size of Flag or Myc-tagged CRY2-mCh is 85 kDa. To check where the band of CRY2 locates in the blot, we prepared three cultures which were transfected with CRY2^{high}, CRY2^{wt} or CRY2^{low} respectively (lane 1-3), and an untransfected control (lane 4). After light illumination, the samples were subjected to Flag-beads pull down and then immunoblotted with anti-Flag or anti-Myc antibodies. In the immunoblots of the total input fraction probed by either Flag or Myc antibody, there is one clear band just above 75 kDa which corresponds to CRY2. A band of the same size is also detected in the IP blots in the transfected samples rather than in non-transfected one, which indicates that Myc-CRY2-mCh can be specifically pulled down. However, in IP blots, there are a number of nonspecific bands, which also appear in untransfected control. For the results presented henceforth, the bands slightly above 75 kDa are extracted from the full blot.

We tried transfection using different amounts of DNA and found that 1.6ug of each DNA per well of a 12-well plate give the best result in the amount of Myc-CRY2 in the IP blot. For all the following transfection, we kept using 1.6ug/per DNA per well of 12-well plate for transfection.

Optimizing the lysis and immunoprecipitation conditions

Inside cells, CRY2-CRY2 homo-interaction is reversible, which means that in the absence of blue light, CRY2 dimer or oligomer would dissociate with a half lifetime ~10 min. To prevent CRY2-CRY2 dissociate during IP, we conducted the 1hr lysis process and 2hr binding with Flag-beads with low level of blue light exposure. We also conducted the IP process in dark or under red light. From the IP blotting results below, similar levels of CRY2-CRY2 interaction were detected either in dark or in light, which indicates that the CRY2 dimer/oligomer is stable during the dark IP process (a condition also used in the previous report of full length CRY2 co-IP⁵). For the following experiments, cell lysis and beads binding steps were conducted in dark or red light.

Optimizing the light illumination duration

We also varied the duration of blue light illumination from 1 min, 5 min, 15 min to 2hr before cell lysis. As shown below, a 5-min blue light illumination is sufficient to induce significant amount of myc-CRY2 co-precipitation.

co-IP Results and conclusion

We used the following experimental conditions: transfection of each well with 1.6ug/each plasmid, blue light illumination for 5 min, IP processing in dark. It is worth noting that during optimization, the three variants of CRY2 did show any obvious trend in the co-precipitation between myc-CRY2 and Flag-CRY2. After optimizing the experimental conditions, we repeated the co-IP experiments three times and the results are shown below (the three repeats showed similar results). With blue light illumination, anti-Myc blot shows that similar amount of Myc-CRY2 was co-precipitated for CRY2^{high}, CRY2^{wt} and CRY2^{low}, when the samples were immune-precipitated with anti-Flag beads. Surprisingly, in dark control without any blue light stimulation, Myc-CRY2 was still co-precipitated among the three variants by Flag-tagged CRY2. In summary, in the co-IP experiments, we did not observe either light-dependent CRY2-CRY2 interaction or the increasing trend of homo-interaction from CRY2^{low}, CRY2^{wt} and CRY2^{high}.

Discussion of the co-IP experiment

We believe that the failed co-IP experiment is likely due to the instability of CRY2 PHR domain *in vitro*. Although the full length CRY2 can be purified *in vitro*, the photolyase homology region (PHR) of CRY2, the protein domain used in our study, cannot be purified *in vitro*. In a recent paper⁶, it was specifically stated that “We were unable to express and purify a shorter variant of CRY2, CRY2PHR, that has also been shown to exhibit light dependent binding to CIB1 in cells.” on the first paragraph, page 55. Therefore, although the co-IP experiment has been convincingly demonstrated for the full length CRY2, it appears not to be a good approach for CRY2 PHR domain.

Though the co-IP experiments failed to show the increasing homo-interaction affinities from CRY2_{low}, CRY2_{wt} to CRY2_{high}, we believe that the western blot results probing CRY2-Raf/ERK activation serves as an alternative, albeit indirect, independent proof of the trend (Fig. 5). Our results showed that CRY2_{low}-Raf, CRY2_{wt}-Raf and CRY2_{high}-Raf induced increasing amount of pERK activation, which indirectly proved the increasingly enhanced affinities of homo-interaction from CRY2_{low}, CRY2_{wt} and CRY2_{high}.

Materials and Methods

2 wells of HEK293T cells in a 12-well plate with 70-80% confluency were co-transfected with Flag-CRY2_{high}-mCh& Myc-CRY2_{high}-mCh, or Flag-CRY2_{wt}-mCh& Myc-CRY2_{wt}-mCh or Flag-CRY2_{low}-mCh& Myc-CRY2_{low}-mCh using Lipofectamine 2000 (Thermo Fisher) and recovered in culture medium for 24hrs. After illumination using home-made LED arrays for specific duration at 0.2mW/cm², cell lysis and IP process were conducted according to previously reported protocol⁵. Briefly, the cells were washed once with PBS and lysed in 3 volumes 1% Brij buffer. After 1 hr incubation on an end-over-end rotator, cell lysates were mixed with 15 μ L anti-Flag beads (Sigma F2426) and incubated with rotation at 4 °C for 2 hr. Beads were pelleted and washed 4 times with cold 1% Brij buffer. Beads were mixed with equal volume of 2x SDS-PAGE Sample Buffer, heated at 100 °C for 4 min, and spun to collect the supernatant for SDS-PAGE analysis. Protein was transferred to a polyvinylidene fluoride membrane using the Trans-Blot Turbo system (Bio-Rad), and then probed using anti-Myc (Covance MMS-150R) or anti-Flag antibody (Sigma F1804). HRP-conjugated secondary antibodies

were used for protein band detection (Cell Signaling Technology 7076).

It has been recently reported that a CRY2 inhibitor BIC1 can suppress CRY2 photodimerization (Science 354:343-347). The authors may want to test how BIC1 affect light responsiveness of CRY2^{High} and CRY2^{low} in at least a control experiment.

--- We have tested whether BIC1 inhibits the light-induced homo-oligomerization of CRY2^{high}, CRY2^{wt} and CRY2^{low}. For this purpose, CRY2 variants were directed to plasma membrane by tagging with CAAX. Membrane bound CRY2 demonstrates increased clustering capacity so that the difference in homo-interaction affinities can be visualized⁷ (for example, CRY2^{wt} and CRY2^{low} do not oligomerize in cytosol). COS7 cells were transfected with CRY2^{high}-mCh-CAAX, CRY2^{wt}-mCh-CAAX or CRY2^{low}-mCh-CAAX respectively, with or without co-expressing GFP-BIC1. A single pulse (100 ms) of blue light at 9.7W/cm² was used to photoexcite CRY2 and CRY2 distribution was examined at t=0s (before blue light) and t=100 s (100s after the blue light pulse). Without CIB1-GFP, CRY2^{high} aggregated dramatically at 100s and formed lots of clusters within a small area, while CRY2^{wt} clustered significantly less (Fig.R3a). CRY2^{low}-mCh-CAAX only formed a few clusters. In cells co-transfected with CRY2-mCh-CAAX and BIC1-GFP, none of the three CRY2 constructs formed noticeable puncta (Fig.R3b). This result shows that BIC1 indeed inhibits light-induced homo-oligomerization of all three CRY2 mutants. This result is presented in supplementary figure 7.

Figure R3 | BIC1 can inhibit light-mediated cluster formation of CRY2^{high}, CRY2^{wt} and CRY2^{low}. COS7 cells were transfected with CRY2^{high}-mCh-CAAX, CRY2^{wt}-mCh-CAAX, and CRY2^{low}-mCh-CAAX respectively (a) or with GFP-BIC1 (b). One pulse blue light of 100 ms exposure was delivered to COS7 cells and CRY2 distribution was acquired at t=0s and t=100 s. (a) The clusters formed increased drastically from CRY2^{low}, CRY2^{wt} and CRY2^{high}. (b) In the presence of GFP-BIC1, none of CRY2^{low}, CRY2^{wt} and CRY2^{high} formed any noticeable cluster. Scale bars, 5 μ m.

Reviewer 3

The light-promoted heterodimerization of the Arabidopsis thaliana photoreceptor cryptochrome 2 (Cry2) with its interacting partner CIB1 underpins many optogenetic applications that rely on light-induced protein-protein interactions. In addition to forming Cry2:CIB1 heterodimers, Cry2 also shows propensity to form higher-order oligomers upon light exposure. For certain applications, this latter activity is desirable but for others it disturbs. An earlier report by Tucker and colleagues achieved a significant step towards understanding and hopefully untangling the concurrent hetero- and homo-association reactions: the mutation E490G, denoted Cry2olig, in the very C terminus of the Cry2 construct routinely used greatly increased clustering capability of Cry2. Cui, Lin and coworkers now follow up and present a suite of Cry2 derivatives that show more or less clustering proficiency while retaining the ability to interact with CIB1 in light-activated manner. This palette of light-gated actuators promises to be useful where exclusively one or the other of the two light-driven responses of Cry2 is called for. The authors construct several N- and C-terminal truncations/substitutions of Cry2, thus altering charge distribution in these regions of the protein; to test homo-/hetero-association, CIB1 is directed to the ER membrane and Cry2 associates in light-dependent manner. Clustering is assessed by puncta formation of fluorescently labeled Cry2. One variant, E490R, shows similar clustering ability as Cry2olig (i.e. E490G), whereas placement of several negative charges in the C terminus reduced clustering. When applied to Raf signaling, the different variants activated the pathway to different extent.

Altogether this is an interesting, largely well written and potentially quite useful contribution. However, there are a couple of aspects that need to be addressed prior to publication. It appears that N-terminal modifications of Cry2 had a marginal effect, especially when compared to the C-terminal modifications. I hence feel that the N-terminal variants could be given short shrift and panels 1b, d, e could be safely removed (or, relegated to Suppl. Mat.) without significant loss to the quality of the manuscript. In a related vein, I suggest toning down on the use of the structural models, both in the

figures and the text. Whereas the initial homology model and its surface charge representation have some merit, I am not convinced by the derivative models that follow (esp. in fig. 1h). The authors just chop off residues in silico but assume that the overall structure of Cry2 is left unaltered; this is much too simplistic an assumption, and I recommend removing altogether these structural models. Moreover, the message the authors want to convey, namely that they add, remove or invert charges, can easily be grasped without these models (except for the original one which is fine).

--- We very much appreciate the reviewer's suggestions to improve the manuscript. We agree with the suggestions and have made several changes. First, we agree that the use of structural models can be tuned down both in the figures and text. We have moved all CRY2 structures (except CRY2wt) to the supplementary materials and also deleted or moved some relevant text to the supplementary information. Second, we agree that details of the N-terminal modifications of CRY2 can be shortened and some information removed. We have moved the original Fig. 1c,d to the supplementary information, and also shortened the relevant text. We only kept original Fig. 1b and e, for the N-terminal studies, which points out the importance of charged residues at CRY2 N-terminus in the light-induced hetero-dimerization of CRY2-CIB1.

Furthermore, I note that throughout the manuscript the Cry2:Cry2 and the Cry2:CIB1 interactions have only been studied in cellular settings which makes it hard to derive precise quantitative conclusions about the efficiency of the various Cry2 species for either reaction. Specifically, differences in intracellular expression, fraction of active protein and trafficking will all factor into the effective intracellular concentration; as a consequence, apparent differences between variants could be solely due to these aspects but need not necessarily reflect a genuine difference in association constants (this issue is nicely discussed in PMID 26137467). Even if real, it would be informative to know in terms of association constants by which factor the variants differ. It is at the editor's discretion to decide on this point but the complete absence of detailed studies with purified Cry2 proteins is a weak point of the manuscript. Despite that, the proof-of-principle experiments indicate that the new Cry2 variants could be quite useful at the practical level (for whatever underlying mechanistic reason); this is particularly true for

the Cry2 variants with reduced clustering proficiency. By contrast, the new variant E490R with more pronounced clustering is only marginally better than the existing Cry2olig. In summary, I believe this manuscript could become suitable for publication in 'Nature Communications' after taking care of these major and the below minor points.

--- We agree with the reviewer about the importance and necessity of *in vitro* experiment using purified CRY2 proteins. However, from the published literature in the last few years, it has been shown that photolyase homology region (PHR) of CRY2 could not be purified *in vitro* and full length cryptochrome 2 behaves differently *in vitro* and *in vivo*.

1. The PHR domain of CRY2, the protein domain used in our study, cannot be purified *in vitro*. The protein has never been previously purified. In a recent paper⁶, it was specifically stated that “We were unable to express and purify a shorter variant of CRY2, CRY2PHR, that has also been shown to exhibit light dependent binding to CIB1 in cells.” on the first paragraph, page 55.

2. The full length CRY2 can be purified *in vitro*. However, unlike in cellular settings, purified full length CRY2 did not show light-dependent association with CIB1. This has been shown in two independent studies^{6, 8}.

3. In the cellular setting, full length CRY2 has different CIB1 affinity and different oligomerization tendency than CRY2PHR, as demonstrated in two previous reports^{9, 10}. This means any measurement using full length CRY2 *in vitro* would not reflect affinity inside cells for CRY2PHR.

To quantify CRY2 oligomerization at different concentrations in the cellular setting

Given these limitations for *in vitro* studies of CRY2PHR (referred as CRY2), we believe the best way is to measure the relative affinity of different CRY2 mutants in cells with a method to measure *in situ* protein concentrations. To this end, we engineered CRY2wt-mCherry-CAAX, CRY2low-mCherry-CAAX and CRY2high-mCherry-CAAX, which are located at the inner leaflet of the plasma membrane. We measured the mCherry

fluorescence intensity (before blue light illumination) for each individual cell, which is used as an indicator of the relative intracellular CRY2 concentration. Then, we converted fluorescence intensities in individual cells to protein concentrations by relating to absolute protein concentrations in western blot. Below is the detailed description of the procedure.

a. Converting intracellular fluorescence intensities to absolute concentrations

The 2-dimensional concentration of CRY2 was calculated by (the total protein amount of CRY2)/(the sum of cell areas).

1. Quantifying the cell covering area per cell culture well

For membrane-bound CRY2, we calculated its 2-dimensional protein concentration using #molecules/area. We first measured the covering area of transfected cells per cell culture well in a 12-well plate. Briefly, 20 mCh images were randomly taken throughout the culture using a 20x objective. The sum of cell areas in each image was measured using ImageJ. The percentage of cell covering area in each image was quantified by (sum of cell areas)/(size of image area). For one well in 12-well plate, the sum of cell areas was calculated by (size of one well in a 12-well plate)*(percentage of cell covering area). In one well, the total area of cells is 103 mm².

2. Quantifying the absolute CRY2 concentration on the cell membrane

We quantified the total amount of Flag-CRY2_{low}-mCh-CAAX protein per cell culture well by using purified mCh-Flag with known concentrations as calibration standards. Briefly, 5 wells of COS7 cells expressing Flag-CRY2_{low}-mCh-CAAX in a 12-well plate were lysed separately using preheated 40μL SDS+β-mercaptoethanol mixture (9:1 ratio). 20 μL (fig.R4a) or 4 μL (fig.R4b) lysate from each lysis was loaded in NuPAGE 4-12% Bis-Tris gel along with varied amount of purified mCh-Flag (0.01μg to 3.2μg) as calibration standards. mCh-Flag was purified from XL10-Gold competent cells expressing mCh-Flag with 6xHis tags at the N-terminus. For quantification, we did Western blot using anti-Flag antibody produced in goat (Santa Cruz Biotechnology) and IRDye 800CW conjugated anti-

goat IgG produced in Donkey (LI-COR Biosciences). Fluorescence from gel was acquired by Odyssey CLx (Fig. R4a: 20 μ L lysates, Fig. R4b: 4 μ L lysates), and band intensities were measured using ImageJ. For both blots, the intensities of three lowest amount (0.01, 0.032, 0.1 μ g mCh-Flag, equivalent to 3.18×10^{-13} , 1.02×10^{-12} , 3.18×10^{-12} mol of proteins) increase linearly as the protein amount increases, which were used to construct the calibration line. The intensities of cell lysates fell within this linear range, and the amount of protein in each cell lysate was calculated using the calibration line. As shown in fig.R4c, the 20 μ l lysate contains 2.91×10^{-12} mol of proteins (in blot shown in Fig. R4a), while the 4 μ l lysate (in blot shown in Fig. R4b) contains about 0.468×10^{-12} mol of Flag-CRY2low-mCh-CAAX (fig.R4d). Therefore, the average amount of Flag-CRY2low-mCh-CAAX protein in one well of 12-well plate (40 μ l lysate) was 5.25×10^{-12} mol, which corresponds to 3.16×10^{12} molecules. Therefore, the average CRY2 concentration is calculated to be 3.16×10^{12} molecules/103 mm² = 3.07×10^4 molecules/ μ m².

Figure R4 | Quantification of CRY2 amount in cell cultures. 5 wells of COS7 cells expressing Flag-CRY2_{low}-mCh-CAAX were lysed in 40 μ L lysate. Gel was loaded with purified mCh-Flag and 20 μ L cell lysates (a) or 4 μ L cell lysates (b) and immunostained against Flag. (c) The band intensity was plotted against protein amount of purified mCh-Flag using results from blot a. (d) The band intensity was plotted against protein amount using purified mCh-Flag using results from blot b. Calibration lines (insets in c and d) were obtained using the intensities of three lowest mCh-Flag amount. The average intensities of cell lysates from 5 wells in either blot were marked in green on the calibration lines.

3. Quantifying the concentration of CRY2 in individual cells.

We quantified the CRY2 concentrations in individual cells by correlating CRY2 protein concentration with mCh fluorescence intensity. We first measured the average fluorescence intensity of Flag-CRY2_{low}-mCh-CAAX. The mCh images of 630 cells expressing Flag-CRY2_{low}-mCh-CAAX were randomly acquired throughout the culture using a 100x objective. The mCh intensity of each cell was then measured using ImageJ and calculated as (average intensity of cell - image background intensity). The average fluorescence intensity of 630 cells was measured as 1900, which would correspond to the average CRY2 concentration/area of 3.07×10^4 molecules/ μm^2 . CRY2 concentrations in individual cells were calculated by linearly scaling with the fluorescence intensities.

b. Assay oligomerization of CRY2_{high}, CRY2_{wt} and CRY2_{low} at various concentrations.

Each of CRY2_{high}-mCh-CAAX, CRY2_{wt}-mCh-CAAX and CRY2_{low}-mCh-CAAX was expressed separately in COS7 cells. mCh fluorescence intensity was measured before blue light stimulation. Then, cells were subject to one pulse of 100 ms blue light at 9.7 w/cm^2 and mCh fluorescence was examined again after 100s (see **Methods** for quantifying cluster mass). Fig.R5a shows three cells with similar expression levels of

CRY2. At $t=100s$, CRY2^{high}-mCh-CAAX aggregated drastically on cell membrane and CRY2^{wt}-mCh-CAAX clustered much less, while CRY2^{low}-mCh-CAAX formed only a few clusters (Fig.R5a also incorporated in supplementary Fig. 7). The cluster mass at $t=100s$ was plotted against the CRY2 concentration (CRY2^{high} $n=48$, CRY2^{wt} $n=48$, CRY2^{low} $n=49$ cells). As shown in Fig.R5b, CRY2^{high}, CRY2^{wt}, and CRY2^{low} have half-maximum clustering concentrations of $4000/\mu m^2$, $13000/\mu m^2$, and $>30000/\mu m^2$ (Fig.R5b also incorporated in Fig. 3 in main text).

Figure R5 | CRY2^{high}, CRY2^{wt} and CRY2^{low} formed different amount of clusters on plasma membrane (a) CRY2^{high}-mCh-CAAX aggregated drastically on cell membrane and CRY2^{wt}-mCh-CAAX clustered much less, while CRY2^{low}-mCh-CAAX formed only a few clusters. (b) CRY2^{high}, CRY2^{wt} and CRY2^{low} formed different amount of clusters at different CRY2 concentrations. (CRY2^{high} $n=48$, CRY2^{wt} $n=48$, CRY2^{low} $n=49$) Scale bars, $5 \mu m$.

C. conclusion

By quantifying homo-oligomerization capacities at various concentrations, we confirmed that indeed, the homo-interacting affinities decrease from CRY2^{high}, CRY2^{wt} to CRY2^{low}.

34/35: does not make much sense to mention ‘the light-oxygen-voltage (LOV) domain’, only to then go on and mention Vivid which also happens to be a LOV protein; presumably the authors mean the paradigmatic *A. sativa* phot1 LOV2 domain and should modify their text accordingly

--- We thank the reviewer for pointing out the ambiguity in this sentence. We agree that we should specify “the light-oxygen-voltage (LOV) domain “ as *Avena sativa* phototropin 1 (AsLOV2) to differentiate it from *Neurospora crassa* vivid. We also included previous work using homo-dimerization of LOV domain and phytochrome, and modified the corresponding introduction as “Optical approaches use photosensory domains that undergo light-induced protein-protein hetero-dimerization, such as LOV (light-oxygen-voltage) domain of *Avena sativa* phototropin 1, *Arabidopsis* phytochrome B and *Arabidopsis* cryptochrome 2, or protein-protein homo-interaction, such as LOV domain of *V. frigida* aureochrome1, *Synechocystis* phytochrome 1 and *Arabidopsis* cryptochrome 2.”

54: mutation underlying Cry2olig should be explicitly named on first occasion

--- Agree. Adjusted the sentence as “CRY2olig, a CRY2 mutant (CRY2(E490G)) with increased oligomerization capacity has been developed”.

69: explicitly state that residues 489-490 are at the C term

--- Agree. Adjusted the sentence as “We also find that electrostatic charges at C-terminal residues 489 and 490 drastically affect light-induced CRY2 homo-oligomerization”.

p6: as discussed above, the apparent differences b/w variants in their interaction with CIB1 might be due to expression etc.-- has this been checked? if so, how? Even so, I don't find the effect all too dramatic and hence deem this section a weaker part of the paper.

--- We are aware that the expression level of CRY2 is a critical parameter for both its homo-oligomerization and hetero-dimerization. To rule out the differences in expression, we specifically selected cells with similar expression levels of CRY2 and

CIB1 (CRY2 fluorescence intensity in 600-900 range, and CIB1 intensity in 4000-5000 range) and average over 10 cells for each mutant. The differences in the CIB1 binding are dramatic for the three mutants. In the main figure, we only presented a subcellular region of a cell. Here below we presented images of multiple cells (Fig. R6). Two seconds after one pulse of blue light (200ms exposure at 9.7 W/cm²), CRY2wt were completely recruited to ER membrane while only a small portion of CRY2(neutral2-6) and CRY2(Δ 2-6) got recruited. Statistical analysis of multiple cells was presented in Supplementary Figure 1c.

Figure R6 | The light-induced CRY2-CIB1 binding is much weaker for CRY2(neutral2-6) or CRY2(Δ 2-6), as compared to CRY2wt. One pulse of 200-ms blue light was delivered to COS7 cells co-transfected with CIB1-GFP-Sec61 and each mCh-tagged CRY2 respectively. Multiple cells were included in the images in the same field of view. Scale bars, 10 μ m.

113: panmultiple cells in the same field of view. el e of fig. 1 is cited before panel d; the same is true for panels g and h further down

--- We have adjusted the figures so that the figure panels are cited in sequence.

123-130: not sure whether the chosen setup is suited to address relative clustering efficiency; rather, to do so, one would need to go to a setting where Cry2 wt is on the verge of clustering to tease out differences of the variants, if any. Could one go back to the setting of the original Bugaj paper where they first reported Cry2 clustering?

--- In the first paper reporting CRY2 clustering as an optogenetic tool¹¹, clustering efficiency was demonstrated by cytosolic CRY2 aggregation. However, another report shows that the CRY2wt only formed a few clusters in very few cells in cytosol⁹. In our studies, we found that CRY2wt indeed formed clusters, but there were usually only a few clusters per cell and not every cell has clusters. Therefore, to test the homo-oligomerization of CRY2 mutants that do not form clusters in cytosol, we recruited CRY2 to intracellular membrane where the clusters form easily.

p7.143: throughout the paper, please refrain from calling the structural models 'structures' -- for what they are worth, they are models and hence should be called thus

--- Thanks. We have made the appropriate corrections in the text and also removed most structural models from Fig. 1.

172: depending on pKa of residue H490, it may be charged or not, so the statement '... two positive charges ...' could potentially be inaccurate. Also, why was lysine (E490K) not tested as well? (seems straight-forward)

--- We thank the reviewer for pointing out the pKa effect. We have adjusted the sentence as "producing two mutants with two positively charged amino acids, CRY2(E490R) and CRY2(E490H)". We chose Arginine and Histidine as positive charges because these two amino acids have very different side chains and very different pKa (pKa=12.10 for Arginine and pKa=6.04). The lysine (pKa=10.67) falls in between. At the reviewer's request, we constructed and tested CRY2(E490K) mutant. Indeed, it formed plenty of clusters in cytosol under the same illumination conditions as

other mutants (Fig. R7). It supports the critical role of C-terminal positively charged amino acids in enhancing CRY2 homo-interactions. This result was included in Supplementary Fig. 4.

Figure R7 | CRY2(E490K) has enhanced clustering capacity. COS7 cells were transfected with CRY2(E490K)-mCh and subject to blue light illumination at 5 s intervals for a total of 500 s. Before any blue light stimulation, CRY2(E490K) distributed homogeneously across the cell. After light exposure, CRY2(E490K) aggregated drastically into clusters. Scale bar, 10 μ m.

p9 middle para: it would be informative to include actual numbers when comparing the variants, rather than saying 'greater cluster mass' or 'similar clustering capabilities' etc.

--- We thank the reviewer for this suggestion. Relevant text has been modified accordingly.

213-218: cf. above, how do we know whether these differences are indeed due to differences in association constants as opposed to conc. differences of active Cry2? at the very least, this should be discussed

--- The CRY2 expression levels for different mutants are similar. The direct evidence is that cells expression different mutants have similar average and spread of fluorescence intensities. For all quantifications, we averaged over many cells. An indirect evidence is the western blot of CRY2^{high}-mCh-Raf, CRY2^{wt}-mCh-Raf, CRY2^{low}-mCh-Raf. We have used anti-Raf antibody to probe the protein expression

levels in bulk. As shown in figure below, the three mutants have similar protein expression levels.

220: reference to Matlab not needed, should be explained in Methods

--- Agree. We have moved the Matlab analysis to the Methods part and Imaging Processing sections.

235: again, what about E490K?

--- We constructed CRY2(E490K) and confirmed its drastically enhanced homo-oligomerization ability (shown in Fig. R7).

287/292: references should presumably be to fig. 5

--- This typo is corrected.

334: fig. 6??

--- This typo is corrected as Fig. 5.

534: '... are removed ...'

--- Corrected.

542: '... amino acid ...'

--- Corrected.

545: '... residues 490 to 487 ...'; does not make much sense, does it?

--- Corrected as 'residues from CRY2(487-490)'.

References

1. Ettinger, A. & Wittmann, T. Fluorescence live cell imaging. *Methods in cell biology* 123, 77-94 (2014).
2. Ong, Q. et al. The Timing of Raf/ERK and AKT Activation in Protecting PC12 Cells against Oxidative Stress. *PLoS One* 11, e0153487 (2016).
3. Zhang, K. et al. Light-mediated kinetic control reveals the temporal effect of the Raf/MEK/ERK pathway in PC12 cell neurite outgrowth. *PLoS One* 9, e92917 (2014).
4. Baker, D. & Sali, A. Protein structure prediction and structural genomics. *Science* 294, 93-96 (2001).
5. Wang, Q. et al. Photoactivation and inactivation of Arabidopsis cryptochrome 2. *Science* 354, 343-347 (2016).
6. Hallett, R.A., Zimmerman, S.P., Yumerefendi, H., Bear, J.E. & Kuhlman, B. Correlating in Vitro and in Vivo Activities of Light-Inducible Dimers: A Cellular Optogenetics Guide. *ACS Synth Biol* 5, 53-64 (2016).
7. Che, D.L., Duan, L.T., Zhang, K. & Cui, B.X. The Dual Characteristics of Light-Induced Cryptochrome 2, Homo-oligomerization and Heterodimerization, for Optogenetic Manipulation in Mammalian Cells. *Acs Synthetic Biology* 4, 1124-1135 (2015).
8. Liu, H.T. et al. Photoexcited CRY2 Interacts with CIB1 to Regulate Transcription and Floral Initiation in Arabidopsis. *Science* 322, 1535-1539 (2008).
9. Taslimi, A. et al. An optimized optogenetic clustering tool for probing protein interaction and function. *Nat Commun* 5, 4925 (2014).
10. Kennedy, M.J. et al. Rapid blue-light-mediated induction of protein interactions in living cells. *Nat Methods* 7, 973-975 (2010).
11. Bugaj, L.J., Choksi, A.T., Mesuda, C.K., Kane, R.S. & Schaffer, D.V. Optogenetic protein clustering and signaling activation in mammalian cells. *Nat Methods* 10, 249-252 (2013).

REVIEWERS' COMMENTS:

Reviewer #2 (Remarks to the Author):

The revision has satisfactorily addressed all the previous concerns of this reviewer, it appears ready for publication

Reviewer #3 (Remarks to the Author):

The authors have taken to heart the reviewers' comments and have gone to great length to address them. Although it is somewhat lamentable (arguably, most disappointing to the authors themselves) that the new coIP experiments proved so difficult, the underlying reason for this outcome is certainly not lack of effort by the authors but rather the well-documented difficulty in handling Cry2 in vitro, also reported by others.

On the whole, I find the manuscript significantly improved and ready for publication in Nat Comm.